# A Time-Reparameterized Cumulative Intensity Extrapolation Sampler for Discrete Flow Matching

**Feiyang Fu** [1]   **Hehe Fan** [1 ✉]

## Abstract

Discrete flow matching (DFM) provides a principled framework for generative modeling on discrete state spaces via continuous-time Markov chain dynamics. In practice, sampling for DFM commonly employs discretizations such as $\tau$-leaping, yet efficient sampling methods under a limited number of function evaluations (NFE) remain less studied. To address this gap, we propose the Time-Reparameterized Cumulative Intensity Extrapolation (TR-CIE) sampler, which aims to improve sampling quality when function evaluations are restricted. TR-CIE consists of two components. First, a schedule-based time reparameterization rescales the time grid according to the noise schedule. Under standard factorized DFM rate parameterizations, this transformation of variables absorbs the schedule-dependent growth term and mitigates stiffness near the terminal sampling stage. Second, we introduce a cumulative-intensity extrapolation updating rule. By reusing cached model outputs from the previous step as a history term, this improves the approximation of stepwise cumulative intensities on the resulting non-uniform time grid. We provide a theoretical analysis that bounds the local approximation error of cumulative intensities and establishes convergence results. The resulting sampler requires one NFE per step and introduces no additional model evaluations compared to the standard $\tau$-leaping sampler. Extensive experiments on synthetic tasks, text generation, and text-to-image benchmarks demonstrate that our method improves sampling quality under limited NFE.

## 1. Introduction

Generative modeling has achieved promising progress in recent years, enabling high-quality synthesis and controllable generation across a wide range of modalities (Kingma & Welling, 2014; Ho et al., 2020; Song et al., 2021; Rombach et al., 2022; Lipman et al., 2023; Geng et al., 2025). Beyond continuous signals such as images and audio, many applications involve inherently discrete representations, including natural language tokens, tokenized images obtained from vector-quantized autoencoders, symbolic sequences, and other categorical data (Van Den Oord et al., 2017; Esser et al., 2021; Stark et al., 2024; Chang et al., 2022). For these domains, it is important to develop generative methods that operate directly on discrete state spaces while maintaining compatibility with large vocabularies and high-dimensional sequences.

Discrete diffusion models (DDMs) provide an established framework for this setting by defining Markovian noising dynamics on a finite state space and learning to reverse them (Austin et al., 2021; Hoogeboom et al., 2021; Campbell et al., 2022; Lou et al., 2024). In continuous-time formulations, the noising process constitutes a continuous-time Markov chain (CTMC), and sampling involves simulating the learned reverse-time CTMC (Campbell et al., 2022). Starting from a prior distribution (such as a uniform distribution or an absorbing state), the learned reverse-time process generates samples by progressively refining the discrete state. This framework has demonstrated strong empirical performance on token-based tasks and has become a standard baseline for discrete generative modeling (Austin et al., 2021; Campbell et al., 2022; Lou et al., 2024).

More recently, discrete flow matching (DFM) has attracted attention as an alternative CTMC-based framework (Gat et al., 2024; Davis et al., 2024; Shaul et al., 2025). Similarly to discrete diffusion models, DFM constructs a CTMC on a discrete state space and learns time-dependent transition rates to transport a simple prior distribution to the data distribution. Unlike discrete diffusion, which typically starts from a predefined forward noising process and derives the reverse-time dynamics, DFM directly specifies a target probability path and trains a generator consistent with that path via a flow matching objective, following the principles

[1]College of Artificial Intelligence, Zhejiang University, Hangzhou, China. Correspondence to: Hehe Fan <hehe-fan@zju.edu.cn>.

*Proceedings of the 43$^{rd}$ International Conference on Machine Learning*, Seoul, South Korea. PMLR 306, 2026. Copyright 2026 by the author(s).

of continuous flow matching (Lipman et al., 2023). This additional flexibility in the selection of probability paths and rate structures can simplify modeling decisions and broaden the design space for training and inference.

### 1.1. Related Work

**Efficient Sampling in Continuous Diffusion and Flow Models** Efforts to accelerate sampling in continuous diffusion and flow models mainly focus on reducing the NFE required to simulate the reverse-time process, which is commonly formulated as an ordinary differential equation (ODE) or a stochastic differential equation (SDE) (Song et al., 2021; Lu et al., 2022; Zhao et al., 2023). One major direction involves developing advanced numerical solvers, including higher-order discretizations, multistep updates, and predictor-corrector methods (Lu et al., 2022; 2025; Liu et al., 2022; Zhao et al., 2023). Another direction optimizes the components of the sampling procedure (Sabour et al., 2024; Xue et al., 2024; Zhou et al., 2024). These methods help to mitigate discretization errors in ODE or SDE solvers for continuous domains. In contrast, sampling for discrete flow matching is driven by a CTMC in a discrete state space, where the significant computational challenge lies in approximating the stepwise cumulative intensity required by $\tau$-leaping.

**Discrete Diffusion Models** DDMs have developed rapidly as a framework for categorical generation, with applications spanning text, tokenized images, graphs, and biological sequences (Austin et al., 2021; Campbell et al., 2022; Hoogeboom et al., 2021; Kim et al., 2025). Several advances in modeling and training have improved the practicality of DDMs. SEDD (Lou et al., 2024) proposes a score entropy loss that extends score matching to discrete spaces. RADD (Ou et al., 2025a) demonstrates that the concrete score in the absorbing diffusion admits an analytic factorization connecting masked diffusion with autoregressive modeling. Recent theoretical studies have also analyzed the convergence properties of DDMs and the impact of score estimation errors on the generated distribution (Wan et al., 2025; Su et al., 2025; Ren et al., 2025a; Zhang et al., 2025; Liang et al., 2025).

Recent work on discrete diffusion models in the absorbing regime has shown that their reverse dynamics admit a time-agnostic first-hitting interpretation (Zheng et al., 2025). From this perspective, the first hitting sampler (FHS) reformulates generation as a token-by-token decoding process by analytically sampling the transition times at which masked tokens are first revealed, rather than relying on a standard time-discretized reverse process. This provides an important alternative view of masked diffusion sampling and is especially natural in the pure absorbing setting.

Another large body of work studies faster inference for DDMs while maintaining sampling quality (Zhu et al., 2025b; Fu et al., 2025; Ren et al., 2025b; Zhu et al., 2025a; Zhao et al., 2025). Approximate simulation methods are widely used to enable parallel updates across multiple dimensions. A standard example is $\tau$-leaping (Gillespie, 2001), which applies an Euler-type approximation to simulate multiple coordinates simultaneously. Tweedie $\tau$-leaping (Campbell et al., 2022) further improves this approximation by leveraging the posterior expectation of the state to correct the transition rates. While these variants are efficient, the inherent discretization bias often necessitates many steps when intensities vary rapidly near the terminal stage. Specifically, the work of Ren et al. (2025b) is highly relevant. This work adapts higher-order numerical methods, namely $\theta$-RK2 and $\theta$-Trapezoidal methods, to discrete diffusion inference. It establishes second-order convergence in KL divergence and typically requires multiple function evaluations per step (e.g., 2 NFE) to form the higher-order corrections. In our method, we introduce a schedule-based time reparameterization to eliminate the schedule-dependent growth term in standard factorized DFM. Furthermore, we propose TR-CIE to estimate stepwise cumulative intensities via history reuse. Our approach maintains a cost of one NFE per step, making it effective for large-scale discrete flow sampling where the number of evaluations is strictly limited.

**Discrete Flow Matching** DFM (Gat et al., 2024; Shaul et al., 2025; Davis et al., 2024; Luo et al., 2025; Cheng et al., 2025; Nisonoff et al., 2025) provides a flexible paradigm for discrete generative modeling by learning CTMC dynamics that transports a simple prior to the data distribution. On the methodology side, Gat et al. (2024) establish a flow matching framework based on CTMC for high-dimensional discrete generation. Shaul et al. (2025) characterize discrete probability paths via kinetic energy minimization. Billera et al. (2025) introduce branching flows to model variable-length sequence transitions. From an information-geometric perspective, Davis et al. (2024) map categorical distributions on the probability simplex to the positive hypersphere via the Fisher–Rao metric to obtain closed-form geodesic paths. Regarding theoretical guarantees, Wan et al. (2025) and Su et al. (2025) relate terminal distribution error to the approximation and estimation errors of the learned intensity field. In terms of applications and architectures (Qin et al., 2025; Li et al., 2025; Navon et al., 2025; Chen et al., 2025; Campbell et al., 2024; Havasi et al., 2025), Qin et al. (2025) apply DFM to graph generation, Wang et al. (2025) adapt pretrained autoregressive multimodal models into a DFM framework, and Ou et al. (2025b) propose discrete neural flow samplers using locally equivariant Transformers.

## 1.2. Contributions

To improve sampling for DFM under limited model evaluations, we propose the Time-Reparameterized Cumulative Intensity Extrapolation (TR-CIE) sampler. Our method targets the stepwise cumulative intensities and improves approximation accuracy through a combination of time reparameterization and history reuse. First, TR-CIE introduces a schedule-based time reparameterization tailored to standard factorized DFM (Gat et al., 2024). This transformation removes the schedule-dependent growth term in the transition intensities and mitigates stiffness near the terminal sampling stage. Second, we introduce a two-evaluation reference estimator to analyze the discretization error of cumulative intensity under the frozen-state approximation, where we then derive a practical one-evaluation variant. This practical estimator reuses cached model outputs from the previous step to extrapolate stepwise cumulative intensities, requiring only one model evaluation per step. We provide a theoretical analysis that relates local approximation errors of the cumulative intensity to the terminal distribution divergence. Our method can be considered as a structured adaptation of known numerical ideas with new design choices for the standard factorized DFM setting. In particular, the time reparameterization uses the factorized rate parameterization to remove the schedule-dependent growth term, and the cumulative-intensity extrapolation is developed in the reparameterized time domain for CTMC sampling. The resulting sampler further yields a practical one-evaluation implementation through cached history reuse. Experiments on text generation, text-to-image benchmarks, and a synthetic countdown task demonstrate that our method achieves significantly improved quality under restricted NFE. Our contributions are summarized as follows:

- We propose TR-CIE, which introduces a schedule-based time reparameterization to mitigate schedule-dependent growth in standard factorized DFM and alleviate stiffness near the terminal sampling stage.

- We develop a two-evaluation reference estimator for theoretical analysis and a practical one-evaluation cumulative intensity extrapolation sampler that reuses cached history to reduce the computational cost.

- Extensive experiments on text generation, text-to-image generation, and a synthetic countdown task across various NFE demonstrate that our method significantly improves sampling quality.

## 1.3. Notation

For any positive integer $N$, let $[N] := \{1, \dots, N\}$ denote the set of integers up to $N$. We distinguish the physical time $t \in [0, 1]$ from the reparameterized time $\tau \in [0, \tau_N]$, and denote the time derivative of the scalar interpolation schedule

$\kappa_t$ by $\dot{\kappa}_t$. The discrete data resides in a state space $\mathcal{X} = \mathcal{S}^D$, where $\mathcal{S} = [V]$ represents the vocabulary of size $V$. Let $(X_t)_{t \in [0,1]}$ denote the CTMC process and use $x_n \in \mathcal{X}$ to refer to a specific state realization at time step $n$. We denote the $d$-th coordinate by $x^d$ and the vector excluding the $d$-th coordinate by $x^{\backslash d}$. Let $\mathbb{I}(\cdot)$ denote the indicator function, $\delta_x(\cdot)$ represent the Kronecker delta distribution concentrated at $x$, and $\mathrm{d}_H(\cdot, \cdot)$ measure the Hamming distance on $\mathcal{X}$.

## 2. Preliminaries

This section reviews the CTMC formulation (Norris, 1998) underlying discrete flow models and introduces the sampling quantities approximated by our method. We focus on time-inhomogeneous CTMC on high-dimensional discrete spaces and specify stepwise cumulative intensities.

### 2.1. CTMC on Finite Discrete Spaces

We consider a CTMC $(X_t)_{t \in [0,1]}$ taking values in a finite discrete state space $\mathcal{X}$.

**Infinitesimal Transition Characterization.** For $x \in \mathcal{X}$ and $y \neq x$, let $u_t(y, x) \geq 0$ denote the transition rate from $x$ to $y$ at time $t$. We define the total exit rate $\lambda_t(x) := \sum_{z \neq x} u_t(z, x)$ and set the diagonal entry $u_t(x, x) := -\lambda_t(x)$, ensuring that $\sum_{y \in \mathcal{X}} u_t(y, x) = 0$ for all $x$. The CTMC is characterized by the short-time expansion as the time increment $h \to 0^+$:

$$\mathbb{P}(X_{t+h} = y \mid X_t = x) = \delta_x(y) + u_t(y, x)\, h + o(h). \quad (1)$$

**Probability flow.** Let $p_t(x) = \mathbb{P}(X_t = x)$ be the marginal probability of state $x$ at time $t$. Then $p_t$ satisfies the Kolmogorov forward equation:

$$\frac{\mathrm{d}}{\mathrm{d}t} p_t(y) = \sum_{x \in \mathcal{X}} u_t(y, x)\, p_t(x). \quad (2)$$

### 2.2. Discrete Flow Matching

DFM learns a time-inhomogeneous CTMC that transports a simple base distribution $p_{\mathrm{src}}$ at $t = 0$ to the target data distribution $p_{\mathrm{data}}$ at $t = 1$. The dynamics are parameterized by a time-dependent generator $u_t^\theta(y, x)$, which defines the induced probability flow by

$$\frac{\mathrm{d}}{\mathrm{d}t} p_t^\theta(y) = \sum_{x \in \mathcal{X}} u_t^\theta(y, x)\, p_t^\theta(x), \qquad y \in \mathcal{X}, \quad (3)$$

where $p_t^\theta$ denotes the marginal distribution induced by the parameterized generator. The generator is subject to the CTMC constraints $u_t^\theta(y, x) \geq 0$ for $y \neq x$ and $\sum_{y \in \mathcal{X}} u_t^\theta(y, x) = 0$ for all $x$. We denote the total exit rate of the model by $\lambda_t^\theta(x) := \sum_{y \neq x} u_t^\theta(y, x)$.

In practice, to scale to high-dimensional discrete data $x \in \mathcal{S}^D$, DFM adopts a sparse coordinate-update parameterization. For each coordinate $d \in [D]$ and token $s \in \mathcal{S} \setminus \{x^d\}$, let $u_t^{d,\theta}(s; x) \geq 0$ denote the rate of replacing the $d$-th coordinate $x^d$ by $s$ conditioned on the current state $x$. The resulting generator can be written as:

$$u_t^\theta(z, x) = \sum_{d=1}^{D} \sum_{s \neq x^d} u_t^{d,\theta}(s; x) \, \mathbb{I}\left(z^{\setminus d} = x^{\setminus d}, \, z^d = s\right).$$
(4)

DFM trains $u_t^\theta$ to match a target probability path that interpolates between $p_{\mathrm{src}}$ and $p_{\mathrm{data}}$ via a simulation-free flow matching objective.

## 2.3. Stepwise Cumulative Intensities

To simulate the learned time-inhomogeneous CTMC, numerical solvers typically operate on a discretized time grid. Let $0 = t_0 < t_1 < \cdots < t_N$ be a time partition, where $t_N < 1$ is a cutoff introduced to avoid numerical singularities, as the learned transition intensities often diverge as $t \to 1$. The key quantity for numerical simulation is the stepwise cumulative intensity, which represents the total transition mass accumulated over a time interval. For a frozen state $x \in \mathcal{X}$ and a channel $(d, s)$ (representing a jump of the $d$-th coordinate to value $s$), we define the cumulative intensity over $[t_n, t_{n+1}]$ as follows:

$$\Lambda_{n,d,s}(x) := \int_{t_n}^{t_{n+1}} u_t^{d,\theta}(s; x) \, \mathrm{d}t, \qquad s \neq x^d. \quad (5)$$

Accordingly, the coordinate-wise total cumulative intensity is given by

$$\Lambda_{n,d}(x) := \sum_{s \neq x^d} \Lambda_{n,d,s}(x) = \int_{t_n}^{t_{n+1}} \lambda_t^{d,\theta}(x) \, \mathrm{d}t, \quad (6)$$

where $\lambda_t^{d,\theta}(x) := \sum_{s \neq x^d} u_t^{d,\theta}(s; x)$. These integrals constitute the numerical targets approximated by various sampling algorithms. Throughout the remainder of this paper, we refer to the transition rate $u_t^{d,\theta}(s; x)$ as the intensity and its integrated value $\Lambda_{n,d,s}(x)$ as the cumulative intensity.

## 2.4. Standard Sampling Algorithms

**Poisson $\tau$-Leaping.** Standard exact simulation methods, such as thinning (Lewis & Shedler, 1979), can be computationally expensive when intensities are high or change rapidly. The $\tau$-leaping method (Gillespie, 2001) offers an efficient approximation by assuming that the intensities remain constant within each time step $[t_n, t_{n+1})$. Using the Euler approximation, the cumulative intensity is estimated as follows:

$$\Lambda_{n,d,s}(x_n) \approx (t_{n+1} - t_n) \, u_{t_n}^{d,\theta}(s; x_n). \quad (7)$$

Here, $u_{t_n}^{d,\theta}(s; x_n)$ denotes the intensity evaluated at the current time step $t_n$ conditioned on the current numerical solution state $x_n$. Based on this approximation, the method draws independent Poisson event counts for each channel:

$$P_{n,d,s} \sim \mathrm{Poisson}\left((t_{n+1} - t_n) \, u_{t_n}^{d,\theta}(s; x_n)\right). \quad (8)$$

These counts $P_{n,d,s}$ represent the number of jumps from $x_n^d$ to $s$ during the interval. A conflict resolution rule is typically applied to handle multiple events within a single coordinate. While efficient, the accuracy of $\tau$-leaping can degrade when intensities vary rapidly within a step, motivating the need for improved estimators of stepwise cumulative intensities.

## 3. Method

Poisson $\tau$-leaping requires approximating the stepwise cumulative intensities that parameterize the channel-wise Poisson event counts. The standard Euler rule can incur significant discretization error under limited NFE, especially for standard DFM parameterizations where a schedule-dependent growth term induces stiffness near the terminal time $t_N$. We address these challenges with TR-CIE, which combines (i) a schedule-based time reparameterization and (ii) a cumulative intensity extrapolation rule that reuses cached model outputs, while using one model evaluation per step. We also describe a two-evaluation reference estimator (TR-CIE-Exact) used primarily for theoretical analysis.

**Time-Reparameterized Sampling Schedule.** Following standard factorized DFM implementations (Gat et al., 2024; Lipman et al., 2024; Su et al., 2025), the time dependence of coordinate-wise intensities is controlled by a scalar schedule $\kappa_t \in [0, 1)$ according to the factorization:

$$u_t^{d,\theta}(s; x) = \frac{\dot{\kappa}_t}{1 - \kappa_t} \, g_t^{d,\theta}(s; x), \qquad s \neq x^d, \quad (9)$$

where $g_t^{d,\theta}(s; x)$ represents the output of the neural network. Even with early stopping at a cutoff $t_N < 1$ (where $\kappa_{t_N} = 1 - \varepsilon$), the schedule-dependent growth term $\frac{\dot{\kappa}_t}{1 - \kappa_t}$ grows rapidly as $t \to 1$. This singularity induces severe stiffness near the terminal stage, rendering numerical integration in the physical time domain unstable.

To address this, we introduce a schedule-based time reparameterization, defining a new time variable $\tau$ as follows:

$$\tau(t) := -\ln(1 - \kappa_t). \quad (10)$$

Under this change of variables, the process remains a CTMC, but its intensities are rescaled by the Jacobian $\frac{\mathrm{d}t}{\mathrm{d}\tau} = \frac{1 - \kappa_t}{\dot{\kappa}_t}$:

$$\tilde{u}_\tau^{d,\theta}(s; x) := u_{t(\tau)}^{d,\theta}(s; x) \frac{\mathrm{d}t}{\mathrm{d}\tau}. \quad (11)$$

Substituting (9) into (11) yields:

$$\tilde{u}_\tau^{d,\theta}(s;x) = g_{t(\tau)}^{d,\theta}(s;x), \qquad s \neq x^d. \qquad (12)$$

Thus, the singular growth term is exactly eliminated in the $\tau$ domain. Consequently, stepwise integration targets the (typically bounded) network output, effectively mitigating stiffness. The logarithmic transformation extends the terminal time to an infinite horizon, requiring a truncation at $t_N$ to define a finite computational window. Unlike standard methods where early stopping is used to avoid numerical instability from diverging intensities, here the reparameterized intensity remains bounded. Additionally, a uniform grid in $\tau$ implicitly creates an adaptive schedule in physical time $t$, which naturally allocates more steps near the terminal stage to handle rapid dynamics without manual tuning.

**TR-CIE** We discretize $[0, \tau_N]$ with a grid $0 = \tau_0 < \tau_1 < \cdots < \tau_N$ and denote $h_n := \tau_{n+1} - \tau_n$ and $r_n := \frac{h_n}{h_{n-1}}$ for $n \geq 1$. At step $n$, under the standard frozen-state approximation used by $\tau$-leaping, the numerical target for each channel $(d, s)$ is the stepwise cumulative intensity:

$$\Lambda_{n,d,s}(x_n) := \int_{\tau_n}^{\tau_{n+1}} \tilde{u}_\sigma^{d,\theta}(s;x_n)\,d\sigma, \qquad s \neq x_n^d. \quad (13)$$

Euler $\tau$-leaping employs the left-endpoint rule $\Lambda_{n,d,s}(x_n) \approx h_n \tilde{u}_{\tau_n}^{d,\theta}(s;x_n)$. Motivated by the traditional Adams-Bashforth methods in numerical analysis (Butcher, 2016), TR-CIE instead approximates (13) by integrating a linear extrapolation of $\tilde{u}_\tau$ constructed from the current model output and a cached output from the previous step, with coefficients adapted to the non-uniform time grid via $r_n$. Specifically, we define

$$\tilde{u}_{n,d,s} := \tilde{u}_{\tau_n}^{d,\theta}(s;x_n), \qquad \tilde{u}_{n-1,d,s} := \tilde{u}_{\tau_{n-1}}^{d,\theta}(s;x_{n-1}),$$

where $\tilde{u}_{n-1,d,s}$ is cached from the previous step. TR-CIE reuses its cached output from channel $(\tau_{n-1}, x_{n-1})$ as a history term and computes

$$\widehat{\Lambda}_{n,d,s} = h_n \left[ \left(1 + \frac{r_n}{2}\right) \tilde{u}_{n,d,s} - \frac{r_n}{2} \tilde{u}_{n-1,d,s} \right]. \quad (14)$$

This update requires one model evaluation at $(\tau_n, x_n)$ and reuses the cached output from $(\tau_{n-1}, x_{n-1})$, introducing no additional model evaluation overhead. Since linear extrapolation does not guarantee non-negativity, we explicitly apply a clamping safeguard to bound the estimated intensity within $[\varepsilon_0, M]$:

$$\widehat{\Lambda}_{n,d,s} \leftarrow h_n \cdot \min\left\{ M, \max\left\{ \varepsilon_0, \frac{\widehat{\Lambda}_{n,d,s}}{h_n} \right\} \right\}, \quad (15)$$

where $\varepsilon_0$ is a small constant to ensure positivity, and $M$ is a large upper bound to maintain numerical stability. We use

this clamped value as the cumulative intensity for Poisson sampling and provide the sampling process in Algorithm 1. For initialization, we set $r_0 = 0$ and $U_{\text{prev}} = 0$, which reduces the first step ($n = 0$) to the Euler rule $\widehat{\Lambda}_{0,d,s} = h_0 \tilde{u}_{\tau_0}^{d,\theta}(s;x_0)$.

**Poisson $\tau$-Leaping Update.** With the estimated cumulative intensities $\widehat{\Lambda}_{n,d,s}$, we proceed with the standard Poisson $\tau$-leaping update. For each dimension $d \in [D]$, we draw independent event counts $P_{n,d,s} \sim \text{Poisson}(\widehat{\Lambda}_{n,d,s})$ for all $s \neq x_n^d$. Let $K_{n,d} := \sum_{s \neq x_n^d} P_{n,d,s}$ denote the total number of jump events in dimension $d$. The state is updated coordinate-wise as follows: if $K_{n,d} = 1$, we set $x_{n+1}^d = s^*$ where $s^*$ is the unique token satisfying $P_{n,d,s^*} = 1$; otherwise (i.e., if $K_{n,d} = 0$ or $K_{n,d} > 1$), the state remains unchanged with $x_{n+1}^d = x_n^d$.

**TR-CIE-Exact** To isolate the discretization error derived purely from temporal integration, we introduce a reference estimator, TR-CIE-Exact. Unlike the practical variant, this estimator computes the history term by evaluating the network at the *current* frozen state $x_n$ but at the *previous* time $\tau_{n-1}$:

$$\tilde{u}_{n-1,d,s}^{(\text{ex})} = \tilde{u}_{\tau_{n-1}}^{d,\theta}(s;x_n). \quad (16)$$

It utilizes the same variable-step extrapolation coefficients:

$$\widehat{\Lambda}_{n,d,s}^{(\text{ex})} = h_n \left[ \left(1 + \frac{r_n}{2}\right) \tilde{u}_{n,d,s} - \frac{r_n}{2} \tilde{u}_{n-1,d,s}^{(\text{ex})} \right]. \quad (17)$$

By fixing the current state, this formulation strictly adheres to the Adams-Bashforth principle, providing a second-order approximation of the cumulative intensity integral $\Lambda_{n,d,s}(x_n)$ with respect to $\tau_n$. Although this approach effectively doubles the computational cost by requiring an additional evaluation at $(\tau_{n-1}, x_n)$, it serves as a precise baseline to quantify the impact of the state drift introduced by reusing cached history in the practical TR-CIE sampler.

## 4. Theoretical Analysis

This section analyzes the approximation error of TR-CIE in the reparameterized time domain, focusing on the estimation of the stepwise cumulative intensity defined in (13). We provide convergence rates for both the theoretical two-evaluation estimator (TR-CIE-Exact) and the practical one-evaluation estimator (TR-CIE). Detailed proofs are provided in Appendix C.

**Assumptions.** This section relies on standard regularity conditions stated in Appendix C.1. (i) the reparameterized intensity is $C^2$ in $\tau$ with bounded derivatives (Assumption C.1). (ii) the target and numerical intensities are bounded within $[\varepsilon_0, M]$ to ensure integrability and stability (Assumption C.2). (iii) the intensity is Lipschitz continuous

---

**Algorithm 1** TR-CIE sampling in the $\tau$ domain

---

**Require:** Estimated intensity $\tilde{u}_\tau^{d,\theta}(s; x)$, time grid $\{\tau_n\}_{n=0}^N$, source $p_{\mathrm{src}}$, floor $\varepsilon_0 > 0$ and optional $M$
**Ensure:** Sample $x_N$
1: $h_n \leftarrow \tau_{n+1} - \tau_n, r_0 \leftarrow 0, r_n \leftarrow \frac{h_n}{h_{n-1}}$ for $n \geq 1$
2: Sample $x_0 \sim p_{\mathrm{src}}$, set $U_{\mathrm{prev}} \leftarrow 0$
3: **for** $n = 0$ to $N - 1$ **do**
4: $\quad U_{\mathrm{cur},d,s} \leftarrow \tilde{u}_{\tau_n}^{d,\theta}(s; x_n)$ for all $d$ and $s \neq x_n^d$
5: $\quad \alpha_n \leftarrow 1 + \frac{r_n}{2}, \beta_n \leftarrow \frac{r_n}{2}$
6: $\quad \widehat{\Lambda}_{n,d,s} \leftarrow h_n(\alpha_n U_{\mathrm{cur},d,s} - \beta_n U_{\mathrm{prev},d,s})$
7: $\quad$ Apply positivity safeguard using (15)
8: $\quad$ Sample $x_{n+1}$ by Poisson $\tau$-leaping using $\widehat{\Lambda}_{n,d,s}$
9: $\quad U_{\mathrm{prev}} \leftarrow U_{\mathrm{cur}}$
10: **end for**
11: **return** $x_N$

---

with respect to the Hamming distance on the state space (Assumption C.3). (iv) the time grid is quasi-uniform with bounded step ratios $r_{\min} \leq r_n \leq r_{\max}$ (Assumption C.4).

### 4.1. Local Cumulative Intensity Error

We first analyze TR-CIE-Exact, which computes the extrapolation using the history term evaluated at the *current* frozen state $x_n$.

**Theorem 4.1.** *Under Assumptions C.1 and C.4, for every step $n \geq 1$ and channel $(d, s)$, the local cumulative intensity approximation error satisfies:*

$$\left|\widehat{\Lambda}_{n,d,s}^{(\mathrm{ex})} - \Lambda_{n,d,s}(x_n)\right| \leq C_1 \cdot h_n^3, \qquad (18)$$

*where $C_1$ is a constant independent of $n$.*

This result validates that the schedule-based extrapolation yields third-order local accuracy ($O(h^3)$) with respect to time when the history term is exact.

Then, we analyze the practical TR-CIE estimator, which reuses the cached model output from the previous step $(\tau_{n-1}, x_{n-1})$ to maintain a cost of one model evaluation per step.

**Theorem 4.2.** *Under Assumptions C.1, C.2, C.3, and C.4, for every step $n \geq 1$ and channel $(d, s)$, the following bound holds along the sampler trajectory:*

$$\begin{aligned}\left|\widehat{\Lambda}_{n,d,s} - \Lambda_{n,d,s}(x_n)\right| &\leq C_2 h_n^3 \\ &+ C_3 h_n \, \mathrm{d}_H(x_n, x_{n-1}),\end{aligned} \qquad (19)$$

*where $C_2, C_3$ are constants independent of $n$.*

**Corollary 4.3.** *Under the conditions of Theorem 4.2, the following bound holds:*

$$\begin{aligned}\mathbb{E}\left[\left|\widehat{\Lambda}_{n,d,s} - \Lambda_{n,d,s}(x_n)\right|\right] &\leq C_2 h_n^3 \\ &+ C_3 h_n \, \mathbb{E}\left[\mathrm{d}_H(x_n, x_{n-1})\right].\end{aligned} \qquad (20)$$

*Moreover, if*

$$\mathbb{E}[\mathrm{d}_H(x_n, x_{n-1})] \leq C_d \, h_{n-1},$$

*for some constant $C_d > 0$ (a condition validated empirically in Appendix D.2), then the expected drift contribution is $O(h_n h_{n-1})$.*

The term $h_n \mathrm{d}_H(x_n, x_{n-1})$ in Theorem 4.2 reflects the error introduced by state changes between steps. Corollary 4.3 highlights that TR-CIE reduces the temporal integration error (the $C_2 h_n^3$ term) by using history-based extrapolation, but introduces a state-mismatch error due to history reuse (the $C_3 h_n \mathbb{E}[\mathrm{d}_H]$ term). We discuss which effect dominates in Remark 4.6.

### 4.2. Global Error Decomposition and Remarks

To relate local errors to the terminal distribution, let $p_{\tau_N}$ denote the terminal distribution of the exact $\tau$-domain CTMC driven by the learned intensities $\tilde{u}_\tau^{d,\theta}$, and let $q_{\tau_N}$ denote the terminal distribution induced by the sampler. A change-of-measure argument (Brémaud, 1981; Campbell et al., 2022) yields an upper bound on the discretization error, which can be categorized as follows:

$$\mathrm{KL}(p_{\tau_N} \| q_{\tau_N}) \leq \mathcal{E}_{\mathrm{freeze}} + \mathcal{E}_{\mathrm{var}} + \mathcal{E}_{\mathrm{int}}. \qquad (21)$$

Here, $\mathcal{E}_{\mathrm{int}}$ aggregates the integration errors of cumulative intensity controlled by Theorems 4.1 and 4.2. $\mathcal{E}_{\mathrm{freeze}}$ captures the pathwise state-freezing approximation error inherent to $\tau$-leaping. $\mathcal{E}_{\mathrm{var}}$ captures the within-step time-variation error at the frozen state. Modeling error between the learned CTMC and the data distribution is orthogonal to this discretization analysis and is not considered here.

*Remark* 4.4 (Trade-off between Efficiency and Accuracy). TR-CIE addresses scenarios with limited model evaluations. While the reference estimator TR-CIE-Exact improves the approximation of cumulative intensities by evaluating the history term at $(\tau_{n-1}, x_n)$, it requires an additional evaluation per step. In contrast, TR-CIE maintains one evaluation per step by reusing history cached at $(\tau_{n-1}, x_{n-1})$. Although this reuse introduces a state drift error, the method provides improved sampling quality compared to Euler $\tau$-leaping under an identical NFE budget.

*Remark* 4.5 (Positivity Safeguards). Linear extrapolation can produce negative intensities. Following established practices in chemical kinetics and discrete diffusion (Ren et al., 2025b; Anderson & Mattingly, 2009; Hu et al., 2011), a clamping safeguard with a floor $\varepsilon_0 > 0$ is applied before sampling Poisson event counts. Appendix D.3 provides empirical evidence that the error induced by this safeguard remains negligible.

*Remark* 4.6 (Temporal Integration and State Drift Error). The error bound in Theorem 4.2 comprises a

temporal integration error $C_2 h_n^3$ and a state drift error $C_3 h_n \mathrm{d}_H(x_n, x_{n-1})$. The integration term is governed by the intensity curvature in the $\tau$ domain, while the drift term arises from state mismatch during history reuse. TR-CIE reduces the integration error via history extrapolation at the cost of introducing the drift component. Empirically (Appendix D.1), the magnitude of the temporal integration error significantly surpasses the state drift error. Consequently, the reduction in integration error dominates the total error, which leads to a net improvement in local accuracy.

## 5. Experiments

This section evaluates the proposed sampler, implemented in the reparameterized time domain. Experiments cover (i) text generation with a DFM backbone (Gat et al., 2024), (ii) text-to-image generation with the FUDOKI backbone (Wang et al., 2025) and (iii) a synthetic countdown task (Zhao et al., 2025). The sampling budget is quantified by the NFE and experiments are conducted on an NVIDIA A800 GPU.

### 5.1. Text Generation

We benchmark inference algorithms on the DFM backbone trained with cross-entropy loss, reporting results for both masked and uniform formulations under a linear scheduler. To ensure a rigorous comparison, all methods are matched by the total NFE. Specifically, for single-stage methods (Euler/Tweedie $\tau$-leaping, TR-CIE), the number of steps equals the NFE. For high-order solvers such as $\theta$-RK2 and $\theta$-Trapezoidal (Ren et al., 2025b), the number of integration steps is halved to maintain the same total NFE. We follow Gat et al. (2024) to report generative perplexity evaluated with GPT-2 on 1024 generated samples (lower is better). Perplexities measured by LLaMA-3 and the unigram entropy metric are reported in Appendix B.

We compare TR-CIE against Euler $\tau$-leaping, Tweedie $\tau$-leaping, $\theta$-RK2 and $\theta$-Trapezoidal methods. For the masked setting, we additionally compare with FHS (Zheng et al., 2025) and its high-order variant, HO-FHS (Extrap.). Table 1 summarizes results for the masked formulation across a range of NFE budgets. Table 2 reports the corresponding evaluation for the uniform formulation without FHS. Across varying budgets, TR-CIE improves sampling quality compared to other advanced sampling methods under matched computation. The improvement is most pronounced in low-NFE regimes, where discretization errors are amplified and the approximation of cumulative intensities becomes a major factor in sampling quality.

### 5.2. Text-to-Image Generation

For text-to-image generation, we evaluate the method on the multimodal FUDOKI backbone (Wang et al., 2025) and

report the accuracy metric from the GenEval (Ghosh et al., 2023) benchmark, which measures text-to-image alignment accuracy (higher is better) over a set of 553 prompts in Table 3. We also report CLIP score (higher is better) on the generated images from FUDOKI in Table 4. All methods utilize the same pre-trained model and probability path configuration. We observe that TR-CIE yields better results across most of the NFE regime than other samplers. The text-to-image results align with the text generation findings, which demonstrate that TR-CIE improves the sampling quality when NFE is limited.

### 5.3. Synthetic Countdown Task

We evaluate the TR-CIE sampler on a synthetic sequence task characterized by strong sequential dependencies, following the setup in (Zhao et al., 2025). The dataset consists of 256-token sequences where tokens take values in $\{0, \ldots, 31\}$ and non-zero tokens must strictly decrease by one. We train a DFM model (Gat et al., 2024) with a Transformer-based probability denoiser using a polynomial scheduler $\kappa_t = t^2$, where we select the cross entropy as the loss function.

Regarding the motivation for time reparameterization with this scheduler, we note that the factorization in Eq. (9) depends on the growth term $\frac{\dot{\kappa}_t}{1 - \kappa_t}$. For $\kappa_t = t^2$, this term becomes $\frac{2t}{1 - t^2}$. As $t \to 1$, the denominator approaches zero and causes the term to diverge. Consequently, the dynamics remain stiff near the terminal time, justifying the necessity of the TR component to smooth the integration landscape for polynomial schedules as well.

We measure performance by the error rate, defined as the proportion of generated samples that violate the countdown rule. As illustrated in Figure 1, TR-CIE achieves lower error rates compared to other baseline samplers across various NFE, which demonstrates that our method effectively preserves the sequential structure during the discrete sampling process.

### 5.4. Ablation Study

We analyze the individual contributions of schedule-based time reparameterization (TR) and cumulative intensity extrapolation (CIE) on the text generation task. We select uniform DFM as the backbone, and the baseline sampler is Euler $\tau$-leaping. As shown in Table 5, we apply TR alone to Euler $\tau$-leaping, which yields a better performance by mitigating near-terminal stiffness. CIE alone also significantly improves the generation quality over the baseline. Nonuniform $t$-grid denotes an Euler $\tau$-leaping baseline on a matched nonuniform physical time grid, obtained by mapping the same uniform $\tau$-grid used by TR-CIE back to physical time $t$ and then applying the original $t$-domain Euler

*Table 1.* Text generation with masked DFM backbone. Metric is GPT-2 perplexity ($\leqslant$) on 1024 samples.

| Sampler | NFE=8 | NFE=16 | NFE=32 | NFE=64 | NFE=128 | NFE=256 |
|---|---|---|---|---|---|---|
| Euler $\tau$-leaping | 495.25 | 270.05 | 189.87 | 177.15 | 164.49 | 151.16 |
| Tweedie $\tau$-leaping | 486.40 | 264.50 | 185.11 | 175.20 | 159.05 | 152.90 |
| $\theta$-Trapezoidal | 589.63 | 261.42 | 180.54 | 171.80 | 150.44 | 139.35 |
| $\theta$-RK2 | 591.30 | 259.25 | 181.92 | 168.45 | 157.59 | 146.10 |
| FHS | 480.80 | 252.10 | 174.20 | 172.60 | 166.80 | 144.88 |
| HO-FHS (Extrap.) | 503.95 | 265.60 | 175.95 | 167.70 | 159.90 | 141.32 |
| TR-CIE | **252.15** | **187.10** | **162.30** | **151.25** | **138.20** | **135.92** |

*Table 2.* Text generation with uniform DFM backbone. Metric is GPT-2 perplexity ($\leqslant$) on 1024 samples.

| Sampler | NFE=8 | NFE=16 | NFE=32 | NFE=64 | NFE=128 | NFE=256 |
|---|---|---|---|---|---|---|
| Euler $\tau$-leaping | 439.94 | 220.20 | 157.10 | 134.47 | 122.04 | 120.24 |
| Tweedie $\tau$-leaping | 435.76 | 216.21 | 151.38 | 134.86 | 121.07 | 117.53 |
| $\theta$-Trapezoidal | 557.20 | 212.87 | 154.42 | 127.62 | 115.00 | 110.28 |
| $\theta$-RK2 | 565.12 | 239.27 | 156.33 | 128.51 | 114.64 | 110.48 |
| TR-CIE | **209.84** | **152.09** | **129.66** | **118.66** | **110.02** | **104.44** |

*Table 3.* Text-to-image generation with FUDOKI backbone on GenEval. Metric is accuracy (%) over 553 prompts.

| Sampler | NFE=4 | NFE=8 | NFE=16 | NFE=32 |
|---|---|---|---|---|
| Euler $\tau$-leaping | 43.85 | 60.67 | 65.31 | 68.21 |
| Tweedie $\tau$-leaping | 44.56 | 61.40 | 64.10 | 67.60 |
| $\theta$-Trapezoidal | 52.45 | 59.03 | 65.80 | 68.71 |
| $\theta$-RK2 | **54.32** | 58.40 | 64.48 | 68.15 |
| TR-CIE | 53.05 | **64.27** | **66.08** | **69.59** |

*Table 4.* CLIP score results for text-to-image generation on 553 text-image pairs from FUDOKI.

| Sampler | NFE=4 | NFE=8 | NFE=16 | NFE=32 |
|---|---|---|---|---|
| Euler $\tau$-leaping | 0.311 | 0.323 | 0.325 | 0.334 |
| Tweedie $\tau$-leaping | 0.307 | 0.323 | 0.324 | 0.325 |
| $\theta$-RK2 | 0.290 | 0.322 | 0.330 | 0.334 |
| $\theta$-Trapezoidal | 0.284 | 0.319 | 0.326 | 0.332 |
| TR-CIE | **0.321** | **0.331** | **0.334** | **0.336** |

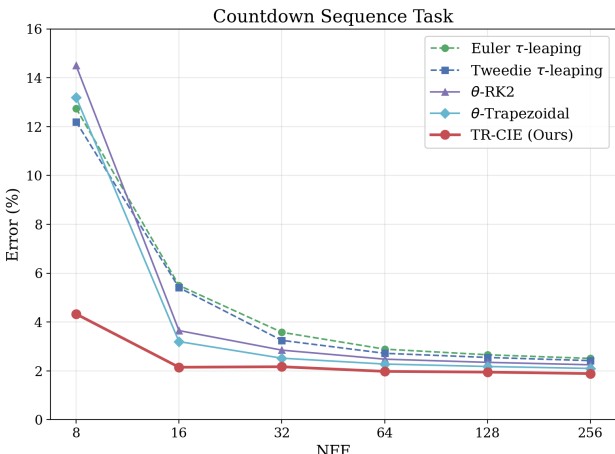

*Figure 1.* **Comparison on the synthetic countdown task.** We compare our TR-CIE sampler with other samplers across various NFE. The error rate (lower is better) indicates that our method generally achieves better performance. We also observe that TR-CIE sampler significantly reduces errors especially in the low-NFE regime.

update. Moreover, the full TR-CIE method achieves the lowest perplexity, which confirms that both components are complementary and essential for better performance under limited NFE.

*Table 5.* Ablation on TR and CIE with uniform DFM backbone. Metric is GPT-2 perplexity ($\leqslant$) on 1024 samples.

| Sampler | NFE=8 | NFE=16 | NFE=32 | NFE=64 | NFE=128 |
|---|---|---|---|---|---|
| Euler | 439.94 | 220.20 | 157.10 | 134.47 | 122.04 |
| TR | 376.06 | 202.91 | 152.35 | 130.88 | 117.85 |
| CIE | 275.87 | 161.34 | 138.62 | 120.97 | 116.78 |
| Nonuniform $t$-grid | 401.32 | 208.47 | 149.86 | 131.59 | 119.45 |
| TR-CIE | **209.84** | **152.09** | **129.66** | **118.66** | **110.02** |

## 6. Conclusion

This paper addresses the challenge of efficient sampling in DFM. Standard approximations such as $\tau$-leaping often suffer from substantial discretization error in this regime. In this work, we present the Time-Reparameterized Cumulative Intensity Extrapolation (TR-CIE) sampler to address these issues. The proposed method utilizes a schedule-based variable transformation to alleviate the stiffness associated with the noise schedule. Additionally, it incorporates an extrapolation mechanism that leverages historical model outputs to refine the estimation of stepwise cumulative in-

tensities. This design improves approximation accuracy while strictly maintaining a cost of one model evaluation per step. Theoretical analysis establishes bounds on the local error and connects these terms to the Kullback-Leibler divergence of the terminal distribution. Empirical evaluations on text generation, text-to-image synthesis, and synthetic sequence tasks demonstrate that TR-CIE produces samples of significantly higher quality than standard baselines under equivalent computational budgets.

## Acknowledgments

This work was supported by the National Natural Science Foundation of China (92570101) and the Earth System Big Data Platform of the School of Earth Sciences, Zhejiang University.

## Impact Statement

This paper presents work whose goal is to advance the field of Machine Learning. There are many potential societal consequences of our work, none of which we feel must be specifically highlighted here.

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

# Appendix

# A Time-Reparameterized Cumulative Intensity Extrapolation Sampler for Discrete Flow Matching

## A. Discussions and Future Work

TR-CIE is designed for standard factorized DFM. In this setting, commonly used schedules, such as linear, quadratic, cubic, and cosine schedules, can induce pronounced terminal stiffness, making time reparameterization particularly effective. For non-factorized parameterizations, or for schedules whose terminal growth is much milder within the sampled horizon, the same cancellation mechanism may not apply directly, and the gains from reparameterization can therefore be smaller.

Our current evidence for image generation is still mainly based on automatic metrics. Conducting larger-scale human evaluations would be an important direction for future work. In addition, it would be an interesting future direction to provide a sharp guarantee that the practical one-evaluation TR-CIE sampler uniformly improves the terminal error over Euler in realistic regimes.

## B. Additional Results

### B.1. More results on text generation

We provide additional results of text generation on both the masked and uniform DFM backbone, where the generative perplexity is measured by LLaMA-3 (Grattafiori et al., 2024). The results are listed in Tables 6 and 7. We observe that our method generally outperforms the other samplers, and the gains are most pronounced at the low NFE regime, for which our method is mainly intended.

We also report unigram entropy on the masked DFM backbone in Table 8. We observe that our method remains broadly comparable to the other samplers, which suggests that the gain in perplexity is not simply coming from a severe reduction in sample diversity.

*Table 6.* Text generation with masked DFM backbone. Metric is LLaMA-3 perplexity ($\leqslant$) on 1024 samples.

| Sampler | NFE=8 | NFE=16 | NFE=32 | NFE=64 | NFE=128 | NFE=256 |
|---|---|---|---|---|---|---|
| Euler $\tau$-leaping | 438.56 | 235.45 | 168.15 | 145.29 | 136.62 | 130.25 |
| Tweedie $\tau$-leaping | 433.12 | 231.85 | 166.50 | 146.06 | 134.25 | 128.81 |
| $\theta$-Trapezoidal | 565.20 | 260.40 | 158.65 | 143.10 | 129.44 | 125.35 |
| $\theta$-RK2 | 579.60 | 255.00 | 162.55 | 144.80 | 131.50 | 125.30 |
| TR-CIE | **231.25** | **167.60** | **140.94** | **131.60** | **120.15** | **118.70** |

*Table 7.* Text generation with uniform DFM backbone. Metric is LLaMA-3 perplexity ($\leqslant$) on 1024 samples.

| Sampler | NFE=8 | NFE=16 | NFE=32 | NFE=64 | NFE=128 | NFE=256 |
|---|---|---|---|---|---|---|
| Euler $\tau$-leaping | 411.00 | 210.25 | 141.15 | 122.19 | 113.62 | 105.25 |
| Tweedie $\tau$-leaping | 410.00 | 208.55 | 142.50 | 121.06 | 110.25 | 103.81 |
| $\theta$-Trapezoidal | 540.20 | 234.10 | 145.65 | 118.90 | 105.94 | 100.55 |
| $\theta$-RK2 | 557.00 | 226.75 | 139.05 | 119.38 | 105.38 | 100.56 |
| TR-CIE | **206.25** | **141.00** | **117.44** | **107.44** | **98.12** | **95.56** |

### B.2. More Results on Image Generation

We conduct image-generation experiments on CIFAR-10 (32×32) using the DFM backbone, following its original setting. We train and evaluate under four schedulers: linear, quadratic, cubic, and cosine. We compare our method with Euler $\tau$-leaping, Tweedie $\tau$-leaping, $\theta$-Trapezoidal and $\theta$-RK2 and report the FID-versus-NFE results in Figure 2. Our method

*Table 8.* Text generation with masked DFM backbone. The metric is unigram entropy on 1024 samples.

| Sampler | NFE=8 | NFE=16 | NFE=32 | NFE=64 | NFE=128 |
|---|---|---|---|---|---|
| Euler $\tau$-leaping | 8.23 | 8.13 | 8.15 | 8.02 | 8.03 |
| Tweedie $\tau$-leaping | 8.23 | 8.14 | 8.13 | 8.03 | 8.02 |
| $\theta$-Trapezoidal | 8.18 | 8.04 | 7.86 | 7.72 | 7.61 |
| $\theta$-RK2 | 8.19 | 8.05 | 7.88 | 7.74 | 7.64 |
| FHS | 8.25 | 8.15 | 8.11 | 8.07 | 8.05 |
| HO-FHS (Extrap.) | 8.24 | 8.11 | 8.10 | 8.07 | 8.04 |
| TR-CIE | 8.17 | 8.10 | 8.05 | 8.03 | 7.92 |

generally outperforms the baselines, especially when NFE is small. Furthermore, we also provide qualitative comparisons at NFE=8 in Figure 4. We observe that our method achieves better semantic alignment and higher generation quality compared with the baseline Euler $\tau$-leaping sampler.

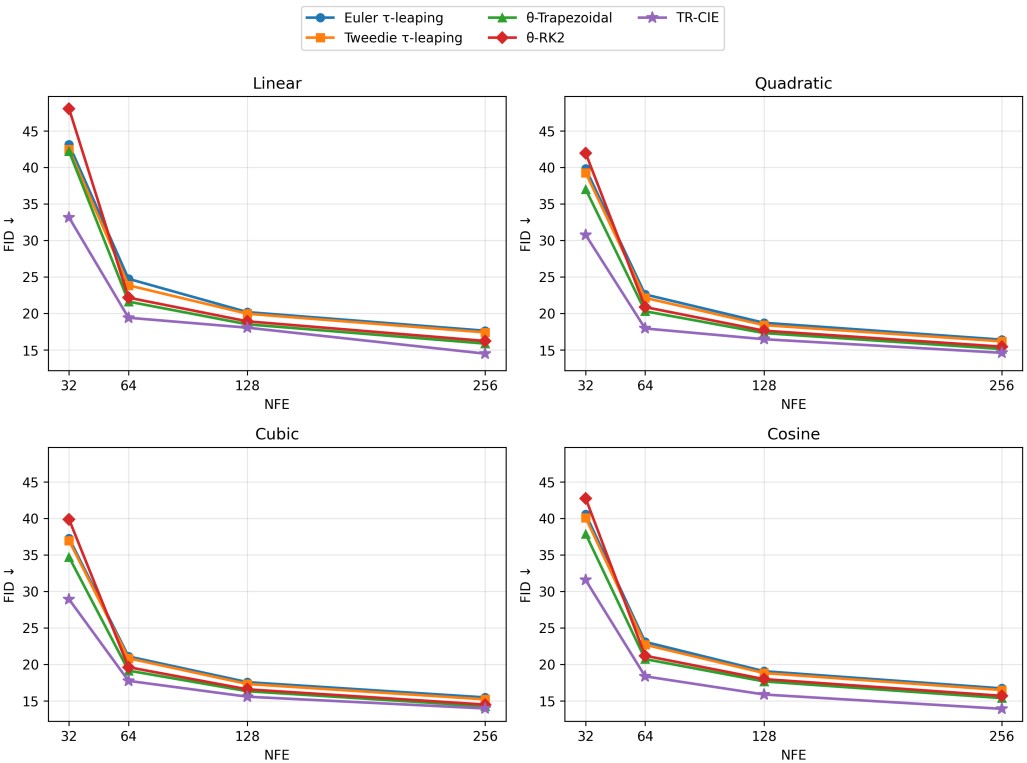

*Figure 2.* **Comparison in terms of FID on the DFM backbone across four sampling schedules.**

### B.3. Runtime comparison

We report the runtime comparison under the same hardware and batch-size settings. The experiments are performed on the uniform DFM backbone using a single NVIDIA RTX 4090 GPU with a batch of 64 samples and report the average generation time per sample at NFE = 8, 16, 32, 64, 128 in Table 9. TR-CIE remains practical in runtime while improving sample quality.

## C. Proofs

This section provides proofs for Theorems 4.1–4.2 and the KL decomposition (21). Throughout, we work in the reparameterized $\tau$-domain and follow the notation established in Sections 2 and 3.

*Table 9.* Runtime per sample across various NFEs.

| Sampler | NFE=8 | NFE=16 | NFE=32 | NFE=64 | NFE=128 |
|---|---|---|---|---|---|
| Euler $\tau$-leaping | 123.8 ms | 242.1 ms | 490.7 ms | 988.9 ms | 1978.6 ms |
| Tweedie $\tau$-leaping | 74.2 ms | 158.2 ms | 313.1 ms | 615.8 ms | 1284.0 ms |
| $\theta$-Trapezoidal | 97.3 ms | 194.1 ms | 388.0 ms | 775.6 ms | 1553.3 ms |
| $\theta$-RK2 | 91.2 ms | 181.9 ms | 366.5 ms | 726.9 ms | 1457.8 ms |
| TR-CIE | 95.5 ms | 190.8 ms | 378.4 ms | 759.7 ms | 1525.4 ms |

### C.1. Assumptions

We recall the $\tau$-domain coordinate-update intensity $\tilde{u}_\tau^{d,\theta}(s;x) \geq 0$ for channels $(d,s)$ with $s \neq x^d$, defined in (11)–(12). To establish the convergence rates, we adopt the following standing assumptions. These conditions are standard in multistep quadrature analysis and change-of-measure arguments for jump processes (Butcher, 2016; Norris, 1998; Ren et al., 2025b; Brémaud, 1981). Specifically, Assumptions C.1 and C.4 support the Taylor-based quadrature bounds. Assumption C.2 ensures the integrability of the log-likelihood ratio in Theorem C.5, and Assumption C.3 controls the additional drift induced by history reuse.

**Assumption C.1** (Time Regularity). For every frozen state $x$ and channel $(d,s)$, the mapping $\tau \mapsto \tilde{u}_\tau^{d,\theta}(s;x)$ is $C^2$ on $[0,\tau_N]$ and satisfies

$$\sup_{\tau \in [0,\tau_N]} \sup_{x,d,s \neq x^d} \left| \partial_\tau^2 \tilde{u}_\tau^{d,\theta}(s;x) \right| \leq M_2. \tag{22}$$

*Justification:* This is a standard smoothness condition for Taylor-based quadrature analysis. We treat it as an idealized regularity assumption to characterize the numerical error.

**Assumption C.2** (Boundedness and Numerical Floor). There exist constants $0 < \varepsilon_0 \leq M < \infty$ such that for all $\tau, d$ and $s$, the target intensity $\tilde{u}$ and the numerical intensity $\hat{\mu}$ satisfy:

$$\varepsilon_0 \leq \tilde{u}_\tau^{d,\theta}(s;x) \leq M, \qquad \varepsilon_0 \leq \hat{\mu}_\tau^d(s) \leq M, \tag{23}$$

where $\hat{\mu}$ denotes the numerical piecewise-constant intensity defined in (25).

*Justification:* The lower bound $\varepsilon_0$ is a standard technical condition in change-of-measure arguments for jump processes to avoid singular log-ratios, and is consistent with applying an explicit positivity floor in implementation. The upper bound $M$ is a boundedness condition for intensity fields, applying to both the model output intensity field and the numerical intensity $\hat{\mu}$.

**Assumption C.3** (State Sensitivity). There exists a constant $L_x > 0$ such that for all $\tau$, channels $(d,s)$, and states $x, x' \in \mathcal{X}$,

$$\left| \tilde{u}_\tau^{d,\theta}(s;x) - \tilde{u}_\tau^{d,\theta}(s;x') \right| \leq L_x \, d_H(x,x'). \tag{24}$$

*Justification:* This is the discrete analogue of the Lipschitz continuity assumption common in continuous diffusion analysis (Song et al., 2021). It posits that a single token change in the input results in a bounded change in the output intensity field.

**Assumption C.4** (Grid Regularity). Let $\{h_n\}$ and $\{r_n\}$ be the step sizes and step ratios defined in Section 3. Assume there exist constants $0 < r_{\min} \leq r_{\max} < \infty$ such that $r_{\min} \leq r_n \leq r_{\max}$ for all $n \geq 1$.

*Justification:* In our experiments, we adopt a uniform grid in the $\tau$-domain (i.e., $h_n \equiv$ const), which implies $r_n = 1$ for all $n$. Thus, this assumption is strictly satisfied with $r_{\min} = r_{\max} = 1$.

### C.2. Pathwise KL Bound

We establish a standard change-of-measure identity for jump processes. This result allows us to upper bound the terminal KL divergence using an integral functional of the intensities and to facilitate the term-by-term error decomposition.

**Target and Numerical Processes** Let $\mathbb{P}$ denote the law of the target CTMC $(X_\tau)_{\tau \in [0, \tau_N]}$ driven by the learned intensities $\tilde{u}_\tau^{d,\theta}(s; X_{\tau-})$. For the numerical process, we define the piecewise-constant numerical intensity $\hat{\mu}_\tau$ for any $\tau \in [\tau_n, \tau_{n+1})$ as follows:

$$\hat{\mu}_\tau^d(s) := \min\left\{ M, \max\left\{ \varepsilon_0, \frac{\widehat{\Lambda}_{n,d,s}}{h_n} \right\} \right\}. \tag{25}$$

Let $\mathbb{Q}$ denote the path measure of the process driven by these intensities. Note that $\hat{\mu}_\tau$ is *predictable* (adapted and left-continuous) as it is determined entirely by the state $X_{\tau_n}$ and history available at the start of the interval $\tau_n$.

**Poisson Bregman Divergence** For scalars $a \geq 0$ and $b > 0$, the Bregman divergence associated with the Poisson likelihood is defined as follows:

$$\mathrm{d}_{\mathrm{Breg}}(a, b) := a \log\left(\frac{a}{b}\right) - a + b. \tag{26}$$

**Theorem C.5** (Pathwise KL Upper Bound). *Under Assumption C.2, the Kullback-Leibler divergence between the terminal distributions is bounded by the expected cumulative Bregman divergence between the intensities:*

$$\mathrm{KL}(p_{\tau_N} \| q_{\tau_N}) \leq \mathrm{KL}(\mathbb{P} \| \mathbb{Q}) = \mathbb{E}_\mathbb{P}\left[ \int_0^{\tau_N} \sum_{d=1}^{D} \sum_{s \neq X_{\tau-}^d} \mathrm{d}_{\mathrm{Breg}}\left( \tilde{u}_\tau^{d,\theta}(s; X_{\tau-}), \hat{\mu}_\tau^d(s) \right) \mathrm{d}\tau \right]. \tag{27}$$

*Proof.* The inequality $\mathrm{KL}(p_{\tau_N} \| q_{\tau_N}) \leq \mathrm{KL}(\mathbb{P} \| \mathbb{Q})$ follows from the Data Processing Inequality (Cover, 1999), as the terminal state is a functional of the full path. The equality is a direct application of the Girsanov theorem for point processes, which expresses the log-likelihood ratio $\log \frac{\mathrm{d}\mathbb{P}}{\mathrm{d}\mathbb{Q}}$ in terms of the cumulative intensities (Wan et al., 2025; Ren et al., 2025b). Assumption C.2 ensures that the intensities are strictly positive and bounded and guarantees the existence of the Radon-Nikodym derivative and the integrability of the Bregman divergence. $\square$

## C.3. TR-CIE-Exact: Local Error

We first analyze the error of the reference estimator. Recall the true stepwise cumulative intensity $\Lambda_{n,d,s}(x)$ defined in (13). The TR-CIE-Exact estimator uses the variable-step extrapolation defined in (17).

**Lemma C.6.** *Under Assumptions C.1 and C.4, for all $n \geq 1$ and channels $(d, s)$, the local error satisfies:*

$$\left| \widehat{\Lambda}_{n,d,s}^{(\mathrm{ex})} - \Lambda_{n,d,s}(x_n) \right| \leq C_{\mathrm{grid}} M_2 h_n^3, \tag{28}$$

*where $C_{\mathrm{grid}} = \frac{1}{6} + \frac{1}{4 r_{\min}}$ is a constant depending only on the grid regularity.*

*Proof.* Fix $x_n, d, s$ and define the frozen-state intensity path $f(\tau) := \tilde{u}_\tau^{d,\theta}(s; x_n)$. Since $f$ is $C^2$ with $|f''| \leq M_2$ (Assumption C.1), we perform Taylor expansions around $\tau_n$:

$$\text{Integral:} \quad \int_{\tau_n}^{\tau_{n+1}} f(\tau)\, \mathrm{d}\tau = h_n f(\tau_n) + \frac{h_n^2}{2} f'(\tau_n) + R_1, \qquad |R_1| \leq \frac{M_2}{6} h_n^3. \tag{29}$$

$$\text{History:} \quad f(\tau_{n-1}) = f(\tau_n) - h_{n-1} f'(\tau_n) + R_2, \qquad |R_2| \leq \frac{M_2}{2} h_{n-1}^2. \tag{30}$$

Substituting (30) into the definition of $\widehat{\Lambda}_{n,d,s}^{(\mathrm{ex})}$ (17) and utilizing $h_{n-1} = \frac{h_n}{r_n}$, we obtain:

$$\widehat{\Lambda}_{n,d,s}^{(\mathrm{ex})} = h_n \left[ \left(1 + \frac{r_n}{2}\right) f(\tau_n) - \frac{r_n}{2}\left( f(\tau_n) - \frac{h_n}{r_n} f'(\tau_n) + R_2 \right) \right]$$

$$= h_n f(\tau_n) + \frac{h_n^2}{2} f'(\tau_n) - \frac{h_n r_n}{2} R_2.$$

Subtracting the true integral expansion (29) from this estimator yields the error:

$$\widehat{\Lambda}_{n,d,s}^{(\mathrm{ex})} - \int_{\tau_n}^{\tau_{n+1}} f(\tau)\, \mathrm{d}\tau = -\frac{h_n r_n}{2} R_2 - R_1.$$

We bound the history remainder term using $h_{n-1} = h_n / r_n$ and $r_n \geq r_{\min}$ (Assumption C.4):

$$\left| \frac{h_n r_n}{2} R_2 \right| \leq \frac{h_n r_n}{2} \cdot \frac{M_2}{2} \left( \frac{h_n}{r_n} \right)^2 = \frac{M_2}{4 r_n} h_n^3 \leq \frac{M_2}{4 r_{\min}} h_n^3.$$

Combining this with the bound for $|R_1|$, we apply the triangle inequality:

$$\left| \widehat{\Lambda}_{n,d,s}^{(\mathrm{ex})} - \Lambda_{n,d,s}(x_n) \right| \leq \left( \frac{1}{6} + \frac{1}{4 r_{\min}} \right) M_2 \, h_n^3,$$

which completes the proof. □

*Proof of Theorem 4.1.* The theorem follows directly from Lemma C.6 by setting $C_1 = C_{\mathrm{grid}} M_2$. □

### C.4. TR-CIE: Local Error with Drift

The practical TR-CIE estimator approximates the history term by reusing the cached value from the previous step $(\tau_{n-1}, x_{n-1})$: the estimator defined in (14).

**Lemma C.7.** *Under Assumptions C.1, C.3, and C.4, for all $n \geq 1$, the local error satisfies:*

$$\left| \widehat{\Lambda}_{n,d,s} - \Lambda_{n,d,s}(x_n) \right| \leq C_{\mathrm{grid}} M_2 h_n^3 + \frac{r_n}{2} h_n L_x \, \mathrm{d}_H(x_n, x_{n-1}). \tag{31}$$

*Proof.* We add and subtract $\tilde{u}_{\tau_{n-1}}^{d,\theta}(s; x_n)$:

$$\begin{aligned}
\widehat{\Lambda}_{n,d,s} &= h_n \left[ \left( 1 + \frac{r_n}{2} \right) \tilde{u}_{\tau_n}^{d,\theta}(s; x_n) - \frac{r_n}{2} \tilde{u}_{\tau_{n-1}}^{d,\theta}(s; x_n) \right] + \frac{r_n}{2} h_n \left( \tilde{u}_{\tau_{n-1}}^{d,\theta}(s; x_n) - \tilde{u}_{\tau_{n-1}}^{d,\theta}(s; x_{n-1}) \right) \\
&= \widehat{\Lambda}_{n,d,s}^{(\mathrm{ex})} + \frac{r_n}{2} h_n \left( \tilde{u}_{\tau_{n-1}}^{d,\theta}(s; x_n) - \tilde{u}_{\tau_{n-1}}^{d,\theta}(s; x_{n-1}) \right).
\end{aligned}$$

By the triangle inequality:

$$\left| \widehat{\Lambda}_{n,d,s} - \Lambda_{n,d,s}(x_n) \right| \leq \left| \widehat{\Lambda}_{n,d,s}^{(\mathrm{ex})} - \Lambda_{n,d,s}(x_n) \right| + \frac{r_n}{2} h_n \left| \tilde{u}_{\tau_{n-1}}^{d,\theta}(s; x_n) - \tilde{u}_{\tau_{n-1}}^{d,\theta}(s; x_{n-1}) \right|.$$

The first term is bounded by Lemma C.6. For the second term, we apply the state sensitivity assumption (Assumption C.3) to the intensity difference:

$$\left| \tilde{u}_{\tau_{n-1}}^{d,\theta}(s; x_n) - \tilde{u}_{\tau_{n-1}}^{d,\theta}(s; x_{n-1}) \right| \leq L_x \, \mathrm{d}_H(x_n, x_{n-1}).$$

Combining these yields the stated bound. □

*Proof of Theorem 4.2.* The theorem follows directly from Lemma C.7. By setting $C_2 = C_{\mathrm{grid}} M_2$ and $C_3 = \frac{r_{\max}}{2} L_x$ (using $r_n \leq r_{\max}$ from Assumption C.4), we recover the form in (19). □

### C.5. KL Decomposition from Intensity Errors

To prove the global error decomposition in Eq. (21), we focus on bounding the expected cumulative Bregman divergence on the RHS of Theorem C.5:

$$\mathrm{RHS} = \mathbb{E}_{\mathbb{P}} \left[ \int_0^{\tau_N} \sum_{d=1}^D \sum_{s \neq X_{\tau^-}^d} \mathrm{d}_{\mathrm{Breg}} \left( \tilde{u}_\tau^{d,\theta}(s; X_{\tau^-}), \hat{\mu}_\tau^d(s) \right) \mathrm{d}\tau \right].$$

Our strategy is to decompose this total integral into stepwise contributions $\sum_{n=0}^{N-1} \mathcal{E}_n$, where each $\mathcal{E}_n$ represents the error accumulated over $[\tau_n, \tau_{n+1})$. We then further decompose each $\mathcal{E}_n$ by inserting intermediate intensity terms.

**Definitions**   Recall the numerical intensity $\hat{\mu}_\tau$ defined in (25). We additionally define the *stepwise average intensity* at the frozen state $X_{\tau_n}$ as follows:

$$\bar{u}_{n,d,s}(X_{\tau_n}) := \frac{1}{h_n} \int_{\tau_n}^{\tau_{n+1}} \tilde{u}_\sigma^{d,\theta}(s; X_{\tau_n})\, d\sigma = \frac{\Lambda_{n,d,s}(X_{\tau_n})}{h_n}. \tag{32}$$

**Lipschitz Property**   Under Assumption C.2, the Bregman divergence is Lipschitz continuous. There exists a constant $C_\Phi$ such that for all relevant intensities $a, a', b, b'$:

$$\left| d_{\mathrm{Breg}}(a,b) - d_{\mathrm{Breg}}(a',b') \right| \le C_\Phi \left( |a-a'| + |b-b'| \right). \tag{33}$$

**Stepwise Decomposition**   Fix a step $n$ and consider $\tau \in [\tau_n, \tau_{n+1})$. The instantaneous error term is $d_{\mathrm{Breg}}(\tilde{u}_\tau(X_{\tau^-}), \hat{\mu}_\tau(\bar{X}_\tau))$, where $\bar{X}_\tau = X_{\tau_n}$ is the state frozen at the start of the step. We apply the Lipschitz property (33) twice to insert intermediate terms:

1. **Introduce Frozen State $X_{\tau_n}$:** Compare the true process $\tilde{u}_\tau(X_{\tau^-})$ with the frozen process $\tilde{u}_\tau(X_{\tau_n})$.

$$d_{\mathrm{Breg}}\left( \tilde{u}_\tau(X_{\tau^-}), \hat{\mu}_\tau \right) \le d_{\mathrm{Breg}}\left( \tilde{u}_\tau(X_{\tau_n}), \hat{\mu}_\tau \right) + C_\Phi \underbrace{\left| \tilde{u}_\tau(X_{\tau^-}) - \tilde{u}_\tau(X_{\tau_n}) \right|}_{\text{State Drift}}. \tag{34}$$

   Here $\hat{\mu}_\tau$ is the predictable numerical intensity defined in (25), which is determined by $X_{\tau_n}$ and is constant on $\tau \in [\tau_n, \tau_{n+1})$.

2. **Introduce Average Intensity $\bar{u}_n$:** Compare the instantaneous frozen intensity $\tilde{u}_\tau(X_{\tau_n})$ with its average $\bar{u}_n(X_{\tau_n})$.

$$d_{\mathrm{Breg}}\left( \tilde{u}_\tau(X_{\tau_n}), \hat{\mu}_\tau \right) \le d_{\mathrm{Breg}}\left( \bar{u}_n(X_{\tau_n}), \hat{\mu}_\tau \right) + C_\Phi \underbrace{\left| \tilde{u}_\tau(X_{\tau_n}) - \bar{u}_n(X_{\tau_n}) \right|}_{\text{Time Variation}}. \tag{35}$$

**Total Error Aggregation**   Substituting (35) into (34), integrating over $\tau \in [\tau_n, \tau_{n+1})$, and taking the expectation $\mathbb{E}_\mathbb{P}$, we obtain the per-step error $\mathcal{E}_n$. Summing over all steps $n = 0 \ldots N-1$ yields the global decomposition:

$$\mathrm{KL}(p_{\tau_N} \| q_{\tau_N}) \le \sum_{n=0}^{N-1} \mathcal{E}_n \le \mathcal{E}_{\mathrm{int}} + \mathcal{E}_{\mathrm{freeze}} + \mathcal{E}_{\mathrm{var}}, \tag{36}$$

where the three components correspond to the three non-zero terms derived above:

- $\mathcal{E}_{\mathrm{int}} = \sum_n \mathbb{E} \int_{\tau_n}^{\tau_{n+1}} d_{\mathrm{Breg}}(\bar{u}_n, \hat{\mu}_\tau) d\tau$;

- $\mathcal{E}_{\mathrm{freeze}} = \sum_n C_\Phi \mathbb{E} \int_{\tau_n}^{\tau_{n+1}} |\tilde{u}_\tau(X_{\tau^-}) - \tilde{u}_\tau(X_{\tau_n})| d\tau$;

- $\mathcal{E}_{\mathrm{var}} = \sum_n C_\Phi \mathbb{E} \int_{\tau_n}^{\tau_{n+1}} |\tilde{u}_\tau(X_{\tau_n}) - \bar{u}_n| d\tau$.

Explicit bounds for these terms are provided in Appendix C.6.

### C.6. Scaling Rates

In this section, we provide the specific bounds for the three error components identified in the global decomposition (21). Let $h_{\max} := \max_n h_n$ denote the maximum step size.

**(i) Freezing Error $\mathcal{E}_{\mathrm{freeze}}$**   This term arises from the state drift $X_{\tau^-}$ relative to the frozen state $X_{\tau_n}$ within a step. Under Assumption C.3, the intensity difference is bounded by the Hamming distance:

$$\left| \tilde{u}_\tau^{d,\theta}(s; X_{\tau^-}) - \tilde{u}_\tau^{d,\theta}(s; X_{\tau_n}) \right| \le L_x\, d_H(X_{\tau^-}, X_{\tau_n}) \le L_x\, N_{(\tau_n, \tau]},$$

where $N_{(\tau_n,\tau]}$ denotes the number of jumps occurring in the interval $(\tau_n, \tau]$. The total exit rate of the system is the sum of intensities over all $D$ coordinates and their $V-1$ possible target values. By Assumption C.2, each intensity entry is bounded by $M$, therefore the total rate is bounded by $D(V-1)M$. Consequently, the expected number of jumps satisfies $\mathbb{E}[N_{(\tau_n,\tau]}] \le D(V-1)M(\tau - \tau_n)$. Integrating this expectation over $\tau \in [\tau_n, \tau_{n+1}]$:

$$\int_{\tau_n}^{\tau_{n+1}} \mathbb{E}[\mathrm{d}_H(X_{\tau^-}, X_{\tau_n})]\,\mathrm{d}\tau \le \int_{\tau_n}^{\tau_{n+1}} D(V-1)M(\tau - \tau_n)\,\mathrm{d}\tau = \frac{D(V-1)M}{2}h_n^2.$$

Summing over all $N$ steps, the global freezing error scales as:

$$\mathcal{E}_{\text{freeze}} = \sum_{n=0}^{N-1} O(h_n^2) = O(h_{\max} \cdot \tau_N) = O(h_{\max}).$$

This confirms that the baseline error inherent to $\tau$-leaping is $O(h)$.

**(ii) Time-Variation Error $\mathcal{E}_{\text{var}}$**   This term captures the deviation of the instantaneous intensity $\tilde{u}_\tau$ from its stepwise average $\bar{u}_n$ at the frozen state. Since $\tilde{u}_\tau$ is $C^2$ in time, we apply Taylor analysis to show that for $\tau \in [\tau_n, \tau_{n+1}]$:

$$|\tilde{u}_\tau(X_{\tau_n}) - \bar{u}_n(X_{\tau_n})| \le C'h_n.$$

Integrating this deviation over the step length $h_n$ yields a local error of $O(h_n^2)$. Summing over all steps, the global time-variation error scales as:

$$\mathcal{E}_{\text{var}} = \sum_{n=0}^{N-1} O(h_n^2) = O(h_{\max}).$$

**(iii) Integration Error $\mathcal{E}_{\text{int}}$**   This term reflects how well our numerical solver approximates the integral $\Lambda_{n,d,s}$. Recall that the numerical intensity $\hat{\mu}_\tau^d(s)$ (defined in (25)) and the average target intensity $\bar{u}_{n,d,s}(X_{\tau_n})$ (defined in (32)) are both constant with respect to $\tau$ within the interval $[\tau_n, \tau_{n+1})$. Therefore, the integral of the Bregman divergence simplifies to the product of the interval length $h_n$ and the divergence value. Using the Lipschitz property of the Bregman divergence:

$$\mathbb{E}\int_{\tau_n}^{\tau_{n+1}} \mathrm{d}_{\text{Breg}}(\bar{u}_n, \hat{\mu}_\tau)\,\mathrm{d}\tau = h_n\,\mathbb{E}\left[\mathrm{d}_{\text{Breg}}\left(\frac{\Lambda_{n,d,s}}{h_n}, \frac{\widehat{\Lambda}_{n,d,s}}{h_n}\right)\right] \lesssim \mathbb{E}|\Lambda_{n,d,s} - \widehat{\Lambda}_{n,d,s}|.$$

Based on our local error theorems, we note that TR-CIE-Exact Theorem 4.1 implies a local error of $O(h_n^3)$. Summing over $N$ steps ($N \propto 1/h$), the global contribution is second-order $O(h_{\max}^2)$. For TR-CIE Theorem 4.2, it implies a local error of $O(h_n^3) + O(h_n \cdot \mathrm{d}_H)$. The first term contributes $O(h_{\max}^2)$, while the second term contributes $O(h_{\max})$. Though TR-CIE maintains the global $O(h_{\max})$ convergence rate, it effectively suppresses the dominant constant in the error bound. Consequently, under a fixed computational budget, this reduction in the error magnitude leads to a better performance compared to the baseline.

# D. Additional Details

In this section, we first provide a quantitative analysis of the error contributions. Then, we perform experiments on the scaling condition of the Hamming Drift. Next, we conduct experiments on the scaling condition of the Hamming Drift, and provide further implementation details regarding grid schedules and positive clamping.

## D.1. Comparison between Temporal Integration and State Drift

To contextualize the state-drift term in Theorem 4.2, we empirically quantify the relative magnitudes of the temporal integration and state drift proxies along sampling trajectories both in the original $t$ domain and the reparamerized $\tau$ domain.

**Experimental Setup**   Following the settings of the main text, we use the masked DFM backbone with NFE=64 and record statistics over 128 trajectories. We define proxies for the two error components, which are the integration term $C_2 h_n^3$ and the drift term $C_3 h_n \mathrm{d}_H$, since they cannot be calculated directly.

**Measurement Definitions**    The temporal integration proxy measures the $\tau$-domain intensity curvature:

$$\text{Curv}(n) := \max_{d,s} \left| \frac{\tilde{u}_{\tau_{n+1}}^{d,\theta}(s; x_n) - 2\tilde{u}_{\tau_n}^{d,\theta}(s; x_n) + \tilde{u}_{\tau_{n-1}}^{d,\theta}(s; x_n)}{h^2} \right|,$$

reflecting the constant $C_2 \propto |\partial_\tau^2 \tilde{u}|$. The state drift proxy measures intensity sensitivity to state changes:

$$\text{Drift}(n) := \max_{d,s} \widehat{L}_x(n) \cdot \text{d}_H(x_n, x_{n-1}), \quad \text{where } \widehat{L}_x(n) \approx \frac{|\Delta \tilde{u}|}{\text{d}_H + \epsilon_{\text{den}}}.$$

These proxies represent the intensity error density introduced by each source per step.

**Dominance of Integration Error and Efficacy of TR**    As shown in the first row of Figure 3, the temporal integration proxy is significantly larger than the drift proxy for the vast majority of the trajectory except for the terminal sampling stage. The second row indicates that the magnitude of the temporal integration proxy uniformly dominates the state drift proxy across all time steps.

Notably, in the $\tau$ domain, the temporal integration proxy decays as the mapped physical $t \to 1$, whereas it explodes in the $t$ domain. This observation demonstrates the efficacy of our time reparameterization in absorbing the schedule stiffness and ensuring smooth, bounded dynamics near the terminal sampling stage. Consequently, reducing the dominant temporal integration error via TR-CIE yields significant improvements in overall sampling accuracy.

### D.2. Hamming Drift Scaling

Corollary 4.3 interprets the drift contribution under the scaling condition $\mathbb{E}[\text{d}_H(x_n, x_{n-1})] \leq C_d h_{n-1}$. To assess this condition empirically, for each NFE budget we estimate the normalized drift ratio

$$R := \max_{n \in \{1, \dots, N-1\}} \frac{\mathbb{E}[\text{d}_H(x_n, x_{n-1})]}{h_{n-1}},$$

where the expectation is approximated by averaging over 128 sampling trajectories. Table 10 reports the resulting $R$ values for the masked DFM backbone. We observe that $R$ varies mildly across various NFE and step size $h_{n-1}$, which indicates the existence of a constant $C_d$.

### D.3. Implementation Details

**$\tau$-Grid Schedule**    As defined in Eq. (10), we map the physical time $t \in [0, 1]$ to the reparameterized time $\tau \in [0, \tau_N]$. In all experiments, we employ a uniform grid in the $\tau$-domain: $\tau_n = n \cdot \left(\frac{\tau_N}{N}\right)$. Due to the logarithmic mapping, this naturally allocates higher resolution near the terminal time $t = 1$ where transition intensities change rapidly.

**Sensitivity sweep on $\epsilon_0$**    We perform a sensitivity sweep on the lower clamp threshold $\epsilon_0$ on the uniform DFM backbone. Table 11 shows that good performance is preserved only when $\epsilon_0$ is kept extremely close to zero; larger values progressively degrade the sampler. This indicates that the clamp should be viewed as a numerical nonnegativity safeguard.

*Remark* D.1 (On the Positivity of Extrapolated Intensities). Due to the nature of linear extrapolation, TR-CIE may occasionally produce negative extrapolated intensities, so our analysis implicitly assumes the extrapolated intensity is nonnegative, as is standard in related multistep corrections for discrete CTMC sampling (Gillespie, 2001; Ren et al., 2025b; Hu et al., 2011; Anderson & Mattingly, 2009). In practice we enforce positivity by clamping before Poisson sampling, and we empirically monitor the clamping frequency on all tasks. We find that positivity holds with relatively high probability, and the clamping frequency decreases as the NFE increases in Table 12. However, clamping is triggered nontrivially in the low-NFE regime, which still leaves some uncertainty about how benign it is in more challenging or out-of-distribution settings.

*Table 10.* **Empirical Hamming-drift scaling for Corollary 4.3.** We report the drift ratio $R$ estimated from 128 sampling trajectories.

|  | NFE = 8 | NFE = 16 | NFE = 32 | NFE = 64 | NFE = 128 | NFE = 256 |
|---|---|---|---|---|---|---|
| $h_{n-1}$ | 0.13863 | 0.06774 | 0.03398 | 0.01627 | 0.00797 | 0.00394 |
| $R$ (mean$\pm$std) | $1018 \pm 58$ | $1046 \pm 49$ | $1071 \pm 38$ | $1083 \pm 41$ | $1112 \pm 46$ | $1125 \pm 44$ |

*Table 11.* **Sensitivity of TR-CIE to the lower clamp threshold $\epsilon_0$.** We report GPT-2 perplexity on 1024 samples for uniform DFM text generation under different NFE.

| $\epsilon_0$ | NFE = 8 | NFE = 16 | NFE = 32 |
|---|---|---|---|
| 0 | 209.84 | 152.09 | 129.66 |
| $1.00 \times 10^{-12}$ | 209.84 | 152.09 | 129.66 |
| $1.00 \times 10^{-10}$ | 214.63 | 159.75 | 132.15 |
| $1.00 \times 10^{-8}$ | 231.88 | 166.30 | 135.15 |
| $1.00 \times 10^{-7}$ | 277.86 | 171.81 | 138.65 |
| $1.00 \times 10^{-6}$ | 2499.68 | 2704.36 | 1858.68 |
| $5.00 \times 10^{-6}$ | 20839.62 | 19125.56 | 16645.76 |

*Table 12.* **Percentage (%) of nonnegative extrapolated intensities across tasks.** We report the fraction of extrapolated $\tau$-domain intensities that are already nonnegative before clamping across different NFE.

| Task | NFE = 8 | NFE = 32 | NFE = 64 | NFE = 128 | NFE = 256 |
|---|---|---|---|---|---|
| Countdown | $97.9 \pm 0.4$ | $98.2 \pm 0.4$ | $98.5 \pm 0.3$ | $98.8 \pm 0.3$ | $99.0 \pm 0.2$ |
| Text-to-image | $95.6 \pm 1.0$ | $96.9 \pm 0.9$ | $98.5 \pm 0.7$ | $99.1 \pm 0.6$ | $99.3 \pm 0.5$ |
| Text (Uniform) | $97.1 \pm 0.7$ | $97.4 \pm 0.6$ | $97.8 \pm 0.5$ | $98.1 \pm 0.4$ | $98.9 \pm 0.3$ |
| Text (Mask) | $95.1 \pm 1.3$ | $95.9 \pm 0.8$ | $96.8 \pm 0.6$ | $97.6 \pm 0.5$ | $98.1 \pm 0.4$ |

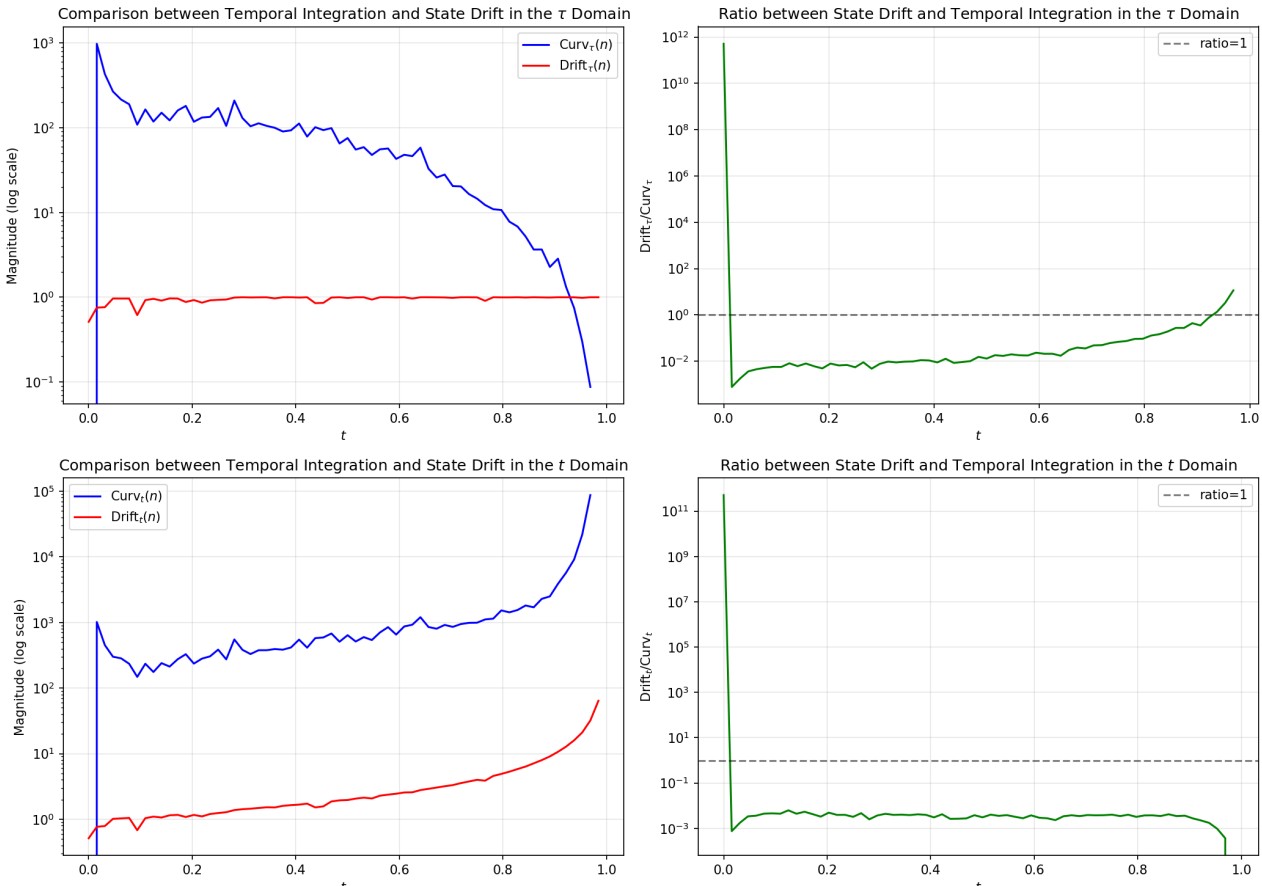

*Figure 3.* **Comparison of temporal integration and state drift proxies. (Bottom)** In the physical $t$ domain, the integration proxy (blue) diverges exponentially as $t \to 1$ due to the schedule-induced stiffness. **(Top)** In the reparameterized $\tau$ domain (mapped back to physical time $t$ for direct comparison in the x-axis), the integration proxy remains bounded and decays near the boundary. The ratio plots (right) show that integration error dominates state drift significantly across most of the trajectory.

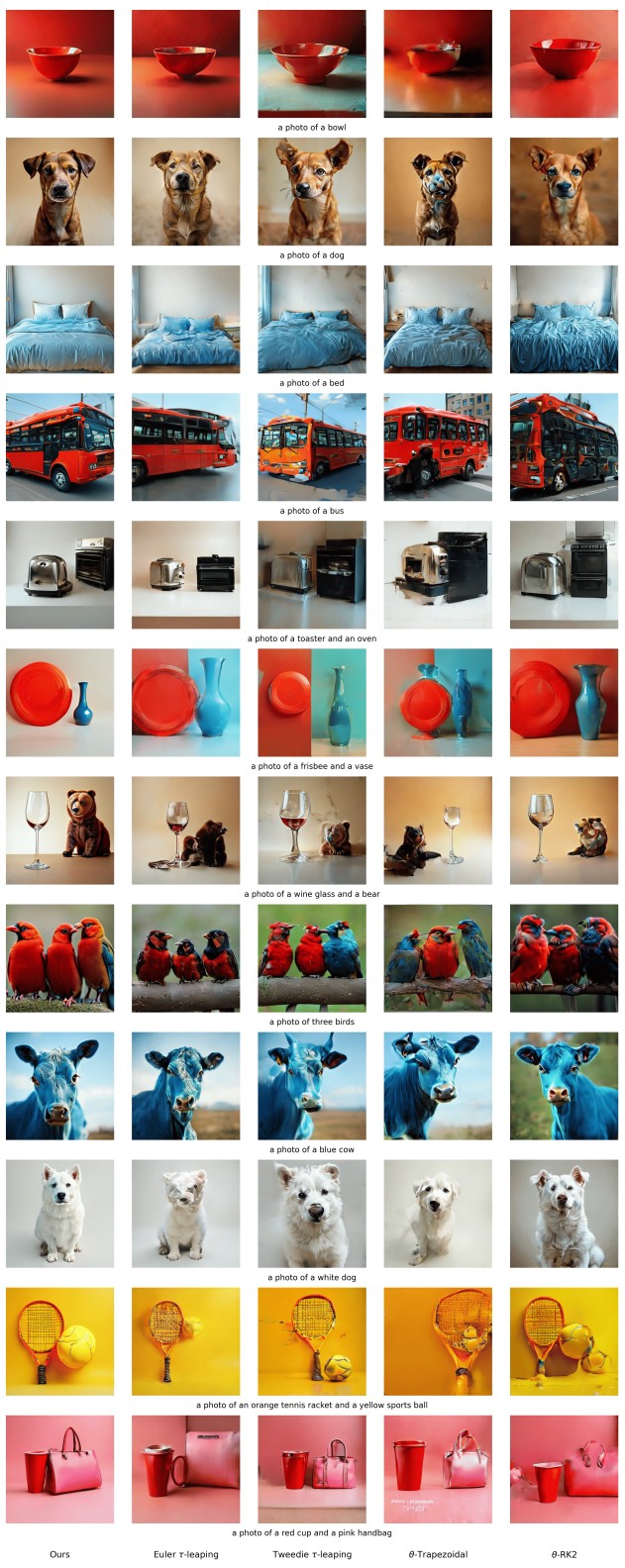

*Figure 4.* **Qualitative comparison at NFE=8.** We observe that TR-CIE better preserves prompt semantics and reduces low-step artifacts, yielding a better sampling quality. (zoom in for best view)

