# OpenReview forum: "A Time-Reparameterized Cumulative Intensity Extrapolation Sampler for Discrete Flow Matching"
_ICML.cc/2026/Conference — ICML 2026 regular_

### Official Review · Reviewer_PEPn · 2026-03-12

**Soundness:** 3
**Presentation:** 3
**Significance:** 3
**Originality:** 3
**Overall Recommendation:** 4
**Confidence:** 3

**Summary:**

This paper proposes TR-CIE, a new sampling algorithm for discrete flow matching that improves generation quality when the number of sampling steps (NFE) is small. The method has two key ideas: 1. time reparameterization: transforms the time variable to remove the growth term in the transition rates. 2. cumulative intensity extrapolation: estimates the cumulative transition intensity using both the current and previous model outputs, improving over the standard euler approximation without increasing the number of model evaluations.

**Compliance With Llm Reviewing Policy:**

Affirmed.

**Final Justification:**

While the rebuttal provides useful clarifications and additional experiments, my main concerns remain. The novelty is still limited, as the method largely reflects a structured adaptation of existing numerical techniques to the DFM setting rather than a fundamentally new approach. The theoretical gap between improved local error and final sample quality is acknowledged but not resolved. Therefore, I maintain my original score.

**Key Questions For Authors:**

1. Relation to existing numerical methods.
The cumulative intensity extrapolation rule appears similar to classical multi-step integration methods (e.g., Adams–Bashforth). Could the authors clarify this connection and explain what aspects are specific to the DFM setting? A clearer discussion would help better assess the novelty of the method.


2. The theoretical analysis focuses on local cumulative intensity error. Can the authors provide further analysis or empirical evidence showing how this improvement translates into better final sampling distributions?


3. The practical method reuses cached outputs from the previous step, introducing a state mismatch term in the error bound. Could the authors provide more analysis on how large this term is in practice and whether it affects stability?


4. Since the motivation is improved sampling efficiency under limited NFE, could the authors provide runtime or latency comparisons with baseline samplers?

**Limitations:**

Yes

**Strengths And Weaknesses:**

**Strengths:**
The paper studies sampling efficiency in DF, particularly in low-NFE regimes, which is relevant for practical discrete generative modeling. The proposed TR-CIE sampler combines time reparameterization with history-based extrapolation while maintaining one model evaluation per step, making it easy to integrate into existing pipelines. The method is well motivated by the stiffness issue near the terminal sampling stage, and the paper provides theoretical analysis showing improved local accuracy of cumulative intensity estimation. Experiments across multiple tasks show improved performance over several baselines, especially when NFE is small.

**Weakness:**
- Limited method novelty. The extrapolation scheme resembles classical multi-step numerical integration methods (e.g., Adams–Bashforth), making the conceptual novelty moderate.
- The analysis mainly provides bounds on local cumulative intensity approximation error, without guarantees on global sampling error or the final sampling distribution.
- Experimental evaluation could be stronger. The paper lacks runtime comparisons and evaluation on larger-scale models.

---

> ### Author Rebuttal · Authors · 2026-03-30
>
> Thank you for your positive assessment and helpful comments. Our responses to the main concerns are as follows.
>
> **(Could the authors clarify this connection and explain what aspects are specific to the DFM setting?)**
>
> Thanks for the question. As stated in the main text, our extrapolation rule is motivated by classical Adams–Bashforth methods. However, it is not a direct application of AB2 to the state dynamics. In ODE solvers, multistep methods extrapolate deterministic derivatives to update a continuous state. In DFM, the state is categorical, and the relevant step quantity is the cumulative intensity that determines the Poisson event counts under the frozen-state approximation. Our extrapolation is therefore applied to this quantity rather than to the discrete state itself, for which a classical multistep update is not directly applicable.
>
> What is specific to the DFM setting is twofold. First, the extrapolation is combined with a time reparameterization that removes the explicit schedule-dependent growth term and mitigates stiffness near the terminal stage. Second, directly applying the original AB2 intuition to cumulative-intensity updating in the CTMC setting would still require two model evaluations to be exact.  We further propose a practical sampler using one evaluation per step by reusing cached history from the previous step.  We will revise the paper to make this positioning more explicit.
>
> **(Can the authors provide further analysis or empirical evidence showing how this improvement translates into better final sampling distributions?)**
>
> Thank you for the comment. We connect the cumulative intensity approximation to the final sampling distribution both theoretically and empirically. On the theory side, Section 4.2 gives a bound on the terminal distribution error that is decomposed into freezing, variation, and integration terms. The integration term is exactly the part affected by the cumulative intensity approximation analyzed in Theorems 4.1 and 4.2. Our claim is therefore not that local error alone determines final sample quality, but that improving the local cumulative intensity approximation reduces one explicit component of the terminal distribution error bound.
>
> More specifically, the two-evaluation reference variant gives the higher-order local integration accuracy, while the practical one-evaluation variant makes the additional state-mismatch term explicit. Empirically, this is reflected in the main text results. In Table 4 of the main text, the ablation shows that adding CIE improves the final sampling metrics under the same NFE budget, and the full TR-CIE method performs best.
>
> **(Could the authors provide more analysis on how large the state mismatch term is in practice and whether it affects stability?)**
>
> Thank you for the comment. We provide both a theoretical bound and empirical evidence for this term. On the theory side, since the practical sampler clamps each channel cumulative intensity by $\hat{\Lambda}\_{n,d,s} \le M h\_n$, the expected one-step state drift is directly controlled as
> $$
> \mathbb{E}\left[d\_H(x\_{n+1},x\_n)\right] \le D(V-1) M h\_n,
> $$
> where $D$ denotes the state dimension and $V$ denotes the vocabulary size. This shows that the additional mismatch caused by reusing cached history decreases with the step size and remains controlled under refinement.
>
> Empirically, Appendix C.2 further examines this term, and the evidence in Table 7 and Figure 3 of the main text is consistent with the same conclusion: the mismatch term decreases as the step size is refined and does not lead to observable instability in the tested regimes. Across text generation, text-to-image generation, and the countdown task, the practical sampler remains stable and improves final sample quality under the same NFE.
>
> **(Could the authors provide runtime or latency comparisons with baseline samplers?)**
>
> Thank you for the suggestion. We additionally report the runtime under the same hardware and batch-size settings. The experiments are performed on the uniform DFM backbone using a single NVIDIA RTX 4090 GPU with a batch of 64 samples. We report the average generation time per sample at NFE = 8 (the runtime at NFE = 16 and 32 grows approximately linearly with NFE). The results are summarized in Table 1 below. Overall, TR-CIE remains practical in runtime while improving sample quality.
>
> **Table 1.** Comparison of runtime on each sample when NFE=8.
>
> | Sampler | NFE=8 |
> |:--|:--:|
> | Euler $\tau$-leaping | 123.8 ms |
> | Tweedie $\tau$-leaping | 74.2 ms |
> | $\theta$-Trapezoidal | 97.3 ms |
> | $\theta$-RK2 | 91.2 ms |
> | TR-CIE | 95.5 ms |

---

> > ### Author Rebuttal · Reviewer_PEPn · 2026-04-03
> >
> > The novelty concern is not resolved, applying extrapolation to cumulative intensity rather than the discrete state is a natural consequence of the DFM setting, not a distinct contribution. The theoretical gap between local error improvement and final sample quality remains open, as your rebuttal effectively acknowledges. The runtime comparison covers only NFE=8, and the concern about larger-scale evaluation is not addressed. Therefore, I keep my score.

---

> > > ### Author Response · Authors · 2026-04-06
> > >
> > > Thanks for your follow-up and for clarifying the remaining concerns. Our responses are given as follows. The additional results mentioned below are provided in [this anonymous link](https://anonymous.4open.science/r/TR-CIE_addtional_responses-7AD3/README.md).
> > >
> > > **(On novelty concern.)**
> > >
> > > Thanks for the comment. We agree that applying extrapolation to cumulative intensity, by itself, is not a distinct contribution. Our point is that the contribution lies in the structured adaptation of known numerical ideas with new design choices for the standard factorized DFM setting. In the reparameterized time domain, cumulative-intensity extrapolation becomes a practical one-evaluation sampler through cached history reuse, while the time reparameterization removes the schedule-dependent growth term under the standard factorized setting. This yields a simple and effective sampler that performs well across several DFM generation tasks, especially in the low NFE regime.
> > >
> > > **(On the theoretical gap between local error improvement and final sample quality.)**
> > >
> > > Thanks for the comment. As stated in our previous rebuttal, local error alone does not determine final sample quality. Here we clarify the connection more precisely. Our theoretical analysis is not limited to purely local error; in Section 4.2, Eq. (21) gives a terminal-distribution KL decomposition, where cumulative-intensity approximation enters through the integration term $\mathcal{E}\_{\mathrm{int}}$. For the present discussion, we do not focus on $\mathcal{E}\_{\mathrm{freeze}}$ and $\mathcal{E}\_{\mathrm{var}}$, since they come from shared freezing and within-step variation mechanisms and are not the main source of difference highlighted by Theorems 4.1 and 4.2.
> > >
> > > The main difference among the considered samplers lies in $\mathcal{E}\_{\mathrm{int}}$. Compared with Euler, whose cumulative-intensity approximation uses a left-endpoint rule, TR-CIE changes this term through a different tradeoff: it reduces the temporal integration part, while introducing an additional state-mismatch term from history reuse. For our practical sampler, $\mathcal{E}\_{\mathrm{int}}$ therefore contains both the temporal integration error and the additional state-drift error from history reuse. Empirically, as shown in Appendix C.1 and Figure 3, the temporal integration proxy is much larger than the drift proxy over most of the trajectory. This suggests that, in the tested regime, improving the dominant part of $\mathcal{E}\_{\mathrm{int}}$ is more beneficial than the additional mismatch cost, which is consistent with the observed improvements in final sampling quality.
> > >
> > > **(On runtime.)**
> > >
> > > Thanks for pointing this out. We additionally report runtime at NFE = 16, 32, 64, and 128 under the same hardware and batch-size settings. The results are provided in Table 1.
> > >
> > > **Table 1.** Runtime across various NFE.
> > >
> > > |Sampler|NFE=16|NFE=32|NFE=64|NFE=128|
> > > |:--|--:|--:|--:|--:|
> > > |Euler $\tau$-leaping|242.1 ms|490.7 ms|988.9 ms|1978.6 ms|
> > > |Tweedie $\tau$-leaping|158.2 ms|313.1 ms|615.8 ms|1284.0 ms|
> > > |$\theta$-Trapezoidal|194.1 ms|388.0 ms|775.6 ms|1553.3 ms|
> > > |$\theta$-RK2|181.9 ms|366.5 ms|726.9 ms|1457.8 ms|
> > > |TR-CIE|190.8 ms|378.4 ms|759.7 ms|1525.4 ms|
> > >
> > >
> > > **(On larger-scale evaluation.)**
> > >
> > > Thanks for the comment. We agree that the original submission did not sufficiently address the larger-scale evaluation concern. For text-to-image, we use the pretrained FUDOKI [1] backbone, which is already a relatively large accessible open DFM-based multimodal model; its original implementation is based on a 1.5B backbone and reports an overall training cost of approximately 43,000 GPU hours. We did find a larger DFM-based omnimodal model, NExT-OMNI, but we could not access its released repository at the time of our check.
> > >
> > > To strengthen the experimental evidence,  we additionally supplement our method with image-generation experiments on the DFM [2] backbone. We follow the original setting of the work and report FID-versus-NFE across four schedules: linear, quadratic, cubic, and cosine. The results can be found in Figure 1 of the link above. While this does not fully resolve the larger-scale evaluation limitation, it broadens the evidence beyond a single pretrained multimodal backbone and a single schedule choice.
> > >
> > > [1] FUDOKI: Discrete Flow-based Unified Understanding and Generation via Kinetic-Optimal Velocities, NeurIPS 2025.
> > >
> > > [2] Discrete Flow Matching, NeurIPS 2024.

---

### Official Review · Reviewer_K48c · 2026-03-13

**Soundness:** 3
**Presentation:** 3
**Significance:** 2
**Originality:** 2
**Overall Recommendation:** 4
**Confidence:** 2

**Summary:**

This paper introduces TR-CIE, a sampler for discrete flow matching (DFM) that pairs two ideas: (i) a time reparameterization that absorbs the schedule-dependent blow-up in the standard factorized rate parameterization, and (ii) a one-step cumulative-intensity extrapolation rule that reuses cached model outputs to improve accuracy without additional NFE. The authors provide local truncation error analysis for both an idealized two-evaluation reference estimator (TR-CIE-Exact) and the practical one-evaluation cached variant, then evaluate on text generation (GPT-2, LLaMA-3), text-to-image generation (GenEval), and a synthetic countdown task. Across reported settings the method consistently outperforms Euler τ-leaping, Tweedie τ-leaping, and several higher-order baselines at low NFE.

**Compliance With Llm Reviewing Policy:**

Affirmed.

**Final Justification:**

Addressed main concerns, so weak accept now.

**Key Questions For Authors:**

- How much of the gain comes from the reparameterization versus generic multistep extrapolation? A comparison against one-NFE extrapolation in the original t-domain, and against a well-tuned nonuniform time grid without the proposed transform, would clarify originality.
- Please report the fraction of clamped channels across all tasks (not only one appendix table) and a sensitivity sweep. If clamping is truly ok, this should be easy to demonstrate.

**Limitations:**

Yes

**Strengths And Weaknesses:**

## Strengths

- **Relevant practical target**. Low-NFE sampling quality is a genuine bottleneck for deploying discrete flow models, and the paper addresses it head-on.
- **Clean reparameterization**. Under the assumed factorized rate form, the time change is mathematically well-motivated and straightforward to implement. The CIE extrapolation is similarly lightweight.
- **Consistent empirical direction**. Results point the same way across all reported tasks, and the ablation (TR alone, CIE alone, TR+CIE) is informative.

## Weaknesses

I actually quite like the paper, though I still have a few open questions. If these are adressed well, I would be willing to raise my evaluation.

- The contribution amounts to a tailored time transform plus an Adams–Bashforth-style history reuse, both specific to the factorized DFM parameterization. This is a sensible numerical engineering improvement, but the scope seems maybe narrower than the general framing suggests. The reparameterization removes the schedule blow-up precisely because of the assumed factorization, and the extrapolation rule is a standard idea transplanted to the reparameterized grid. I would like to see the authors engage more explicitly with how much of this is genuinely new versus an adaptation of known ODE solver techniques to a particular DFM structure.
- The cleanest error bound is for TR-CIE-Exact (two evaluations). The deployed one-evaluation method inherits an additional state-drift term and the argument that this is small rests on empirical validation in the appendix rather than a controlled bound. The global KL decomposition separates freezing, variation, and integration contributions but does not deliver a sharp end-to-end guarantee that the practical sampler improves terminal error over Euler in realistic regimes. The paper occasionally implies stronger theoretical backing than the results actually provide.
- Extrapolated intensities can go negative and are clamped into [ε₀, M], if I understand correctly. eg at NFE = 8 the appendix reports only ~95.1% of intensities are nonnegative, so the safeguard fires nontrivially in exactly the low-NFE regime the paper targets. No sensitivity analysis of ε₀ and M is provided, and the paper does not establish that clamping is benign beyond an assertion that empirical error is small.

Besides this, the evaluation is a bit limited. Text evaluation uses GPT-2 perplexity on 1024 samples, the image evaluation reports only GenEval accuracy on 553 prompts plus qualitative examples; the third task is a toy countdown dataset.

---

> ### Author Rebuttal · Authors · 2026-03-30
>
> Thanks for your comprehensive comments and questions. Due to the space limit, please refer to all additional tables in the [anonymous link](https://anonymous.4open.science/r/TRCIE_ICML2026_Response-2231/README.md). Our responses to the main concerns are as follows.
>
> **(W1: How much of this is genuinely new versus an adaptation of known ODE solver techniques to a particular DFM structure.)**
>
> Thanks for the question. Our method is indeed relevant to Adams–Bashforth methods, which we stated in the paper. However, it is not a direct application of AB2 style extrapolation to the state dynamics. In ODE solvers, multistep methods extrapolate deterministic derivatives to update a continuous state. In DFM, the state is categorical, and the relevant step quantity is the cumulative intensity that determines the Poisson event counts under the frozen-state approximation. Directly applying the original AB2 intuition to cumulative-intensity updating in the CTMC setting would still require two model evaluations to be exact. Our contribution is therefore the combination of time reparameterization, cumulative-intensity extrapolation, and a practical one-evaluation approximation. We will revise the paper to make this positioning more explicit.
>
> **(W2: The global KL decomposition separates freezing, variation, and integration contributions but does not deliver a sharp end-to-end guarantee that the practical sampler improves terminal error over Euler in realistic regimes.)**
>
> Thanks for the comment. We agree that the cleanest local guarantee is for the two-evaluation reference variant. For the practical one-evaluation sampler, the additional state-mismatch term can still be controlled from the deployed algorithm itself.  Since each intensity entry is bounded by M, we have $\hat{\Lambda}\_{n,d,s} \le M h\_n$, which implies
> $$
> \mathbb{E}[d\_H(x\_{n+1}, x\_n)] \le D(V-1) M h\_n .
> $$
> Substituting this into Corollary 4.3 gives
> $$
> \mathbb{E}\left[\left|\hat{\Lambda}\_{n,d,s}-\Lambda\_{n,d,s}(x\_n)\right|\right]
> \le C\_2 h\_n^3 + C\_3 D(V-1) M h\_n h_{n-1}.
> $$
> This yields a local error of order $O(h_n^2)$ and a global integration contribution $E_{\mathrm{int}} = O(h_{\max})$. Combined with the bounds $E_{\mathrm{freeze}} = O(h_{\max})$ and $E_{\mathrm{var}} = O(h_{\max})$ from Appendix B.6, this gives a first-order end-to-end bound:
> $$
> \mathrm{KL}(p\_{\tau\_N}\|q\_{\tau\_N}^{\mathrm{TR\text{-}CIE}}) \le C h\_{\max}.
> $$
> That said, we agree that this does not yet amount to a sharp theorem showing that the practical sampler generally improves terminal KL over Euler in realistic regimes. We will revise the paper to state this scope more carefully.
>
> **(W4: The evaluation is a bit limited.)**
>
> Thanks for the comment. We add diversity evaluation for text generation and new baselines; please refer to Table 4 in the anonymous link.
>
> **(Q1: How much of the gain comes from the reparameterization versus generic multistep extrapolation?)**
>
> Thanks for the question. This is already partially addressed by our ablation in Table 4 in the main text, where CIE (w/o TR) corresponds to one-NFE extrapolation in the original $t$-domain, while Euler (TR) isolates the effect of reparameterization alone. Following your suggestion, we further compare Euler (TR) against Euler on a matched nonuniform $t$-grid without TR, obtained by mapping the same uniform $\tau$-grid used by TR-CIE back to physical time $t$ and then applying the original $t$-domain Euler update. Results in Table 1 of the link show that this baseline improves over uniform-grid Euler but remains worse than Euler (TR), while TR-CIE further improves over Euler (TR). This indicates that the gain is not explained by generic one-NFE extrapolation or better step allocation alone.
>
> **(W3 & Q2: Please report the fraction of clamped channels across all tasks and a sensitivity sweep.)**
>
> Thanks for the suggestion. We add the fraction of clamped channels across all tasks and a sensitivity sweep on the uniform DFM backbone. The upper threshold $M$ is set sufficiently large and is not active in our experiments. Table 2 in the link shows that most extrapolated channels are already nonnegative before correction, with the positive fraction above 95% even at low NFE and increasing with NFE, while Table 3 shows that good performance is preserved only when $\epsilon_0$ is kept extremely close to zero; larger values progressively degrade the sampler. This indicates that the clamp should be viewed as a numerical nonnegativity safeguard. Enforcing positivity after higher-order extrapolation is also standard in related numerical samplers [1,2,3].
>
> [1] Fast Solvers for Discrete Diffusion Models: Theory and Applications of High-Order Algorithms, NeurIPS 2025
>
> [2] A weak trapezoidal method for a class of stochastic differential equations. Communications in Mathematical Sciences, 9(1):301–318, 2011.
>
> [3] A weak second order tau-leaping method for chemical kinetic systems. The Journal of Chemical Physics, 135(2), 2011.

---

> > ### Author Rebuttal · Reviewer_K48c · 2026-04-06
> >
> > Thank you for the detailed rebuttal and for adding additional experiments and clarifications.
> >
> > Overall, I find that several of my concerns have been meaningfully addressed. However, a few concerns remain:
> >
> > - The additional derivation showing a first-order global bound is helpful, but it still does not fully establish that the one-evaluation TR-CIE variant consistently improves terminal error over Euler in realistic regimes. I appreciate that the authors now acknowledge this more explicitly, but this remains a gap between the empirical claims and the theoretical guarantees.
> > - While the additional tables indicate that most intensities are already nonnegative and that performance is stable for small epsilon, the fact that clamping is triggered nontrivially in the low-NFE regime still leaves some uncertainty about how benign it is in more challenging or out-of-distribution settings. But fine :)
> > - The rebuttal improves the positioning relative to classical multistep methods, but I still feel the paper would benefit from a slightly more conservative framing of the contribution as a structured adaptation of known numerical ideas to the factorized DFM setting, rather than suggesting a more broadly novel class of samplers.
> >
> > If the authors have any final comments on these, I'd love to hear them.

---

> > > ### Author Response · Authors · 2026-04-06
> > >
> > > Thank you for the comments and for your time to read our rebuttals. Our additional responses are given as follows.
> > >
> > > **(On the gap between the empirical claims and the theoretical guarantees.)**
> > >
> > > Thanks for the comment. We agree that our current theory does not establish a sharp theorem showing that the practical one-evaluation TR-CIE sampler uniformly improves terminal error over Euler in realistic regimes, which we leave for future exploration. We will revise the paper to make this more explicit.
> > >
> > > Here we would like to provide only a heuristic explanation for why the empirical gains are plausible. In Eq. (21), the method-specific difference between Euler and TR-CIE lies in $\mathcal{E}\_{\mathrm{int}}$. For Euler on the same $\tau$-grid, the local cumulative-intensity approximation error is first order in the quadrature sense, with
> > > $$
> > > \hat{\Lambda}^{\mathrm{Euler}}\_{n,d,s}=h\_n \tilde{u}^{d,\theta}\_{\tau_n}(s;x\_n),
> > > \qquad
> > > \mathbb{E} \left[\left|\hat{\Lambda}^{\mathrm{Euler}}\_{n,d,s}-\Lambda\_{n,d,s}(x\_n)\right|\right]\le C\_E h\_n^2.
> > > $$
> > > For practical TR-CIE, the previous rebuttal gives
> > > $$
> > > \mathbb{E}\left[\left|\hat{\Lambda}\_{n,d,s}-\Lambda\_{n,d,s}(x\_n)\right|\right]
> > > \le C\_2 h\_n^3 + C\_3 D(V-1) M h\_n h\_{n-1}.
> > > $$
> > >
> > > Thus, we do not claim a uniformly smaller bound than Euler. Rather, TR-CIE trades a smaller temporal integration term for an additional state-drift term. Empirically, Remark 4.6 and Figure 3 in Appendix C.1 show that the temporal integration proxy is much larger than the drift proxy for most of the trajectory, suggesting that reducing the temporal integration term is more beneficial than the additional mismatch cost. This is consistent with the observed empirical gains.
> > >
> > > **(On the clamping issue.)**
> > >
> > > Thanks for the comment. We agree that “clamping is triggered nontrivially in the low-NFE regime still leaves some uncertainty about how benign it is in more challenging or out-of-distribution settings”. We will revise the paper to acknowledge this point explicitly.
> > >
> > > **(On the framing of the contribution.)**
> > >
> > > We appreciate this suggestion and agree that a more conservative framing is appropriate. We will revise the Contributions part of the paper to clarify that our contribution is not a broadly new class of samplers. The following content will be added:
> > >
> > > Our method can be considered as a structured adaptation of known numerical ideas with new design choices for the standard factorized DFM setting. In particular, the time reparameterization uses the factorized rate parameterization to remove the schedule-dependent growth term, and the cumulative-intensity extrapolation is developed in the reparameterized domain for CTMC sampling. The resulting sampler further yields a practical one-evaluation implementation through cached history reuse. Our method is easy to integrate and empirically effective in the low-NFE regime through this DFM-specific adaptation.

---

### Official Review · Reviewer_geEY · 2026-03-13

**Soundness:** 3
**Presentation:** 3
**Significance:** 2
**Originality:** 2
**Overall Recommendation:** 5
**Confidence:** 4

**Summary:**

The paper proposes the Time-Reparameterized Cumulative Intensity Extrapolation (TR-CIE) sampler to accelerate sampling of discrete flow matching. First, a schedule-based time reparameterization rescales the time grid according to the noise schedule. Under standard factorized DFM rate parameterizations, this transformation of variables absorbs the schedule-dependent growth term and mitigates stiffness near the terminal sampling stage. Second, a cumulative-intensity extrapolation updating rule is introduced. By reusing cached model outputs from the previous step as a history term, this improves the approximation of stepwise cumulative intensities on the resulting non-uniform time grid. Extensive experiments demonstrate that the method improves sampling quality under limited NFE.

**Compliance With Llm Reviewing Policy:**

Affirmed.

**Final Justification:**

The added baselines and explanations are convincing, and they resolved my concern about the FHS sampler and entropy issue. I appreciate the generalizability to CTMC sampling beyond the absorbing/masked case.

**Key Questions For Authors:**

See weaknesses.

**Limitations:**

No limitation is discussed

**Strengths And Weaknesses:**

Strengths:
- The paper is clearly written and well structured.
- The paper provides meaningful theory, analyzing cumulative-intensity approximation error and relating it to terminal distribution error.
- The experiments are broad, including text generation, text-to-image, a synthetic countdown task, and both masked and uniform DFM backbones.

Weaknesses:
- Most critically, the masked-case evaluation is incomplete. In the absorbing/masked regime, it is known that the original sampling process is theoretically equivalent to a token-by-token decoding process by a first-hitting sampler [1], and time discretization is not needed. The paper only compares with time-discretization-based samplers, while omitting FHS and its high-order variants.
- The text-generation evaluation reports only generative perplexity on 1024 samples, without entropy or another diversity metric. This is a serious issue because [1] shows that inaccurate Gumbel sampling can artificially improve generative perplexity while reducing diversity, making perplexity alone an incomplete metric, especially for time-discretization-based samplers. The authors seem to be unaware of this.
- The paper does not sufficiently position itself against the masked/absorbing line of work. Its related-work section emphasizes discretization-based samplers for discrete diffusion, but does not acknowledge the result that masked diffusion is time-agnostic and can be sampled through the first-hitting view, which directly weakens the novelty claim in that regime.
- The practical method also relies on clamping extrapolated intensities to enforce positivity, so the final sampler is more heuristic than the clean derivation may suggest. The current theory does not fully resolve the effect of this modification on end-to-end sample quality.

[1] Masked Diffusion Models are Secretly Time-Agnostic Masked Models and Exploit Inaccurate Categorical Sampling

---

> ### Author Rebuttal · Authors · 2026-03-30
>
> Thanks for your useful comments and questions. Our responses to the main concerns are given as follows. All citations refer to the reference list provided by the author.
> **(The paper only compares with time-discretization-based samplers, while omitting FHS and its high-order variants.)**
>
> Thanks for pointing this out. We agree that FHS [1] and its higher-order variants are relevant baselines, and we have added them under the same evaluation setup and matched NFE budget. The experiments in Table 1 are performed using the masked DFM [2] backbone.
>
> **Table 1.**  Text generation results. We report GPT-2 perplexity/unigram entropy on 1024 samples.
> |Sampler|NFE=8|NFE=16|NFE=32|NFE=64|NFE=128|
> |:--|:--:|:--:|:--:|:--:|:--:|
> |Euler $\tau$-leaping|495.25/8.23|270.05/8.13|189.87/8.15|177.15/8.02|164.49/8.03|
> |Tweedie $\tau$-leaping|486.40/8.23|264.50/8.14|185.11/8.13|175.20/8.03|159.05/8.02|
> |$\theta$-Trapezoidal|589.63/8.18|261.42/8.04|180.54/7.86|171.80/7.72|150.44/7.61|
> |$\theta$-RK2|591.30/8.19|259.25/8.05|181.92/7.88|168.45/7.74|157.59/7.64|
> |FHS|480.80/8.25|252.10/8.15|174.20/8.11|172.60/8.07|166.80/8.05|
> |HO-FHS (Extrap.)|503.95/8.24|265.60/8.11|175.95/8.10|167.70/8.07|159.90/8.04|
> |TR-CIE|252.15/8.17|187.10/8.10|162.30/8.05|151.25/8.03|138.20/7.92|
>
> We observe that under the same NFE, our method achieves the best overall perplexity while not leading to a significant entropy drop. We will include these results in the revised paper.
>
> **(The text-generation evaluation reports only generative perplexity on 1024 samples, without entropy or another diversity metric.)**
> Thanks for your comment. We additionally computed unigram entropy for the text-generation experiments following the settings in the work for DFM[2].
>
> As shown in Table 1, the perplexity improvement of TR-CIE is not accompanied by a large entropy drop. The entropy remains broadly comparable to the other samplers, which suggests that the gain in perplexity is not simply coming from a severe reduction in sample diversity.
>
> **(The paper does not sufficiently position itself against the masked/absorbing line of work.)**
>
> Thanks for the comment. We will add a paragraph to discuss the masked or absorbing line of work more explicitly, especially the time agnostic perspective. The paragraph to be added is as follows:
>
> Recent work on discrete diffusion models in the absorbing regime has shown that their reverse dynamics admit a time agnostic first hitting interpretation [1]. From this perspective, the first hitting sampler (FHS) reformulates generation as a token by token decoding process by analytically sampling the transition times at which masked tokens are first revealed, rather than relying on a standard time discretized reverse process. This provides an important alternative view of masked diffusion sampling and is especially natural in the pure absorbing setting.
>
> Our work instead focuses on improving sampling quality within the intensity based DFM framework under a limited NFE budget. Concretely, we improve the approximation of the stepwise cumulative intensities used in CTMC sampling. This question remains relevant in few step sampling and also extends beyond the pure absorbing regime, where the first hitting formulation is not directly available.
>
> **(The practical method also relies on clamping extrapolated intensities to enforce positivity, so the final sampler is more heuristic than the clean derivation may suggest.)**
>
> Thanks for the comment. We provide additional empirical evidence to clarify the role of the positivity safeguard. Specifically, we report the fraction of extrapolated intensities that are already nonnegative before clamping across all tasks; due to the space limit, please refer to [this url](https://anonymous.4open.science/r/TR-CIE-Response-E9D5/README.md) for the results. We observe that most extrapolated intensities are valid even before correction, and the need for correction decreases as NFE increases.
> This does not appear to be a method-specific issue. In discrete-state samplers with extrapolation, intermediate estimates can become negative even when the true rates are nonnegative, so positivity correction is a standard practical safeguard [3][4][5]. We will revise the paper to state this more clearly: the theory supports the extrapolation mechanism, while clamping is an implementation-level safeguard for enforcing nonnegativity. We will also explicitly acknowledge this as a limitation.
>
> [1] Masked Diffusion Models are Secretly Time-Agnostic Masked Models and Exploit Inaccurate Categorical Sampling, ICLR 2025.
>
> [2] Discrete Flow Matching, NeurIPS 2024.
>
> [3] Fast Solvers for Discrete Diffusion Models: Theory and Applications of High-Order Algorithms, NeurIPS 2025
>
> [4] A weak trapezoidal method for a class of stochastic differential equations. Communications in Mathematical Sciences, 9(1):301–318, 2011.
>
> [5] A weak second order tau-leaping method for chemical kinetic systems. The Journal of chemical physics, 135(2), 2011.

---

> > ### Author Rebuttal · Reviewer_geEY · 2026-04-04
> >
> > Thanks for the detailed responses. The added baselines and explanations are convincing. I appreciate the generalizability to CTMC sampling beyond the absorbing/masked case. I will raise my score.

---

> > > ### Author Response · Authors · 2026-04-06
> > >
> > > Thank you for the follow-up and for carefully reading our rebuttal. We sincerely appreciate your positive feedback and are glad that the additional baselines and clarifications addressed your concerns.

---

### Official Review · Reviewer_813Z · 2026-03-14

**Soundness:** 3
**Presentation:** 3
**Significance:** 3
**Originality:** 3
**Overall Recommendation:** 4
**Confidence:** 4

**Summary:**

This paper tackles low-NFE sampling for discrete flow matching and argues that standard samplers suffer from stiffness near the terminal stage and poor cumulative-intensity approximation. The proposed TR-CIE sampler combines a schedule-based time reparameterization with a cached-history extrapolation rule while still keeping one model evaluation per step.

**Compliance With Llm Reviewing Policy:**

Affirmed.

**Key Questions For Authors:**

see weakness

**Limitations:**

No. The paper should say more clearly that the gains are concentrated in low-NFE settings and that the evidence on image quality is still fairly metric-driven.

**Strengths And Weaknesses:**

Pro:
- The paper is focused and goes directly after a specific problem that matters for discrete flow matching.
- The method preserves the same one-NFE-per-step budget, which makes the comparison fair and practical.
- The evaluation is broader than one toy benchmark, covering text generation, GenEval text-to-image in Table 3, and the countdown task in Figure 2.


Con:
- The quality metrics are still fairly indirect; GPT-2 perplexity and GenEval are useful, but I wanted a stronger perceptual or human check on the image side.
- The gains shrink as NFE grows, so the contribution is mainly about a specific operating regime rather than a universally better sampler.
- The theory is more convincing for the time-reparameterization motivation than for the exact extrapolation rule.
- The paper could do a better job discussing when TR-CIE would fail, especially for schedules that do not induce the same stiffness pattern.

---

> ### Author Rebuttal · Authors · 2026-03-30
>
> Thank you for your positive assessment and helpful comments. Our responses to the main concerns are as follows.
>
> **(The quality metrics are still fairly indirect; GPT-2 perplexity and GenEval are useful, but I wanted a stronger perceptual or human check on the image side.)**
>
> We agree that stronger perceptual evidence on the image side would strengthen the empirical study. We therefore additionally report CLIP score and include a small scale human preference evaluation. Specifically, at NFE = 4, we randomly sample 100 prompts and ask three annotators to choose the best image among the five methods for each prompt. We report the overall selection frequency aggregated over all 300 votes. The results are shown in Table 1.
>
> **Table 1.** CLIP score and human preference results for text-to-image generation. We report CLIP score at different NFEs and human preference on 100 randomly sampled prompts at NFE = 4, aggregated over 3 annotators (300 total votes).
>
> | Sampler | NFE=4 | NFE=8 | NFE=16 | NFE=32 | Human Preference (%) ↑ |
> |:--|--:|--:|--:|--:|:--:|
> | Euler $\tau$-leaping | 0.311 | 0.323 | 0.325 | 0.334 |18.0 |
> | Tweedie $\tau$-leaping | 0.307 | 0.323 | 0.324 | 0.325 |14.0 |
> | $\theta$-RK2 | 0.290 | 0.322 | 0.330 | 0.334 | 17.0 |
> | $\theta$-Trapezoidal | 0.284 | 0.319 | 0.326 | 0.332 |15.0 |
> | TR-CIE | 0.321 | 0.331 | 0.334 | 0.336 |36.0 |
>
> **(The gains shrink as NFE grows, so the contribution is mainly about a specific operating regime rather than a universally better sampler.)**
>
> Thanks for the comment. The largest gains indeed appear in the limited NFE regime, which is the intended operating regime of the paper. We will revise to claim that our goal is not to develop a uniformly better sampler at arbitrarily large NFE, but to improve sampling quality when the number of model evaluations is constrained.
>
> This behavior is also consistent with the method design. TR-CIE targets the discretization error in the stepwise cumulative intensities, which is most significant when the step size is large. As NFE increases, all samplers become more accurate, the gap naturally narrows.
>
> **(The theory is more convincing for the time-reparameterization motivation than for the exact extrapolation rule.)**
>
> Thanks for the comment. The theoretical motivation is indeed most direct for the time reparameterization, since it follows analytically from the standard factorized DFM form. For the extrapolation rule, the derivation is a variable step extrapolation of the cumulative intensity in the reparameterized time domain, which is the quantity used by CTMC simulation under the frozen state approximation. In this sense, the connection to classical multistep integration is at the level of cumulative intensity approximation rather than state update.
>
> The two evaluation reference variant gives the cleanest local accuracy guarantee for this construction, while the practical one evaluation rule introduces an additional state mismatch term, which is made explicit in Theorem 4.2 and Corollary 4.3.
>
> **(The paper could do a better job discussing when TR-CIE would fail, especially for schedules that do not induce the same stiffness pattern.)**
>
> Thanks for the suggestion. We agree that this scope should be stated more clearly. TR-CIE is designed for standard factorized DFM, which is also the setting considered in the prior work we build on [1,2,3]. In this setting, common schedules such as linear, quadratic, cubic, and cosine induce pronounced terminal stiffness, so the time reparameterization is effective.
> For non-factorized parameterizations, or for schedules whose terminal growth is much milder over the sampled horizon, the same cancellation does not apply directly, and the benefit from reparameterization may be smaller. We will clarify this scope and discuss such failure cases more explicitly in the revision.
>
> **(The paper should say more clearly that the gains are concentrated in low-NFE settings and that the evidence on image quality is still fairly metric-driven.)**
>
> Thanks for the comment. We will make this point explicit in the revision. Our method is mainly intended for the low NFE regime, where the gains are most pronounced. We will also clarify that the current evidence on image quality is still primarily metric-based, while now adding CLIP score and a small-scale human preference evaluation.
>
> [1] Discrete Flow Matching. NeurIPS 2024
>
> [2] Flow Matching Guide and Code. arXiv preprint arXiv:2412.06264, 2024.
>
> [3] A Theoretical Analysis of Discrete Flow Matching Generative Models. arXiv preprint arXiv:2509.22623, 2025.

---

> > ### Author Rebuttal · Reviewer_813Z · 2026-04-03
> >
> > Thanks for the rebuttal. I have carefully read the authors’ response as well as the opinions of the other reviewers.
> >
> > However, the authors did not explain how the “small-scale human preference evaluation” was constructed or conducted. In addition, the noise schedule is a critical component in this field, as its design typically depends on factors such as image resolution, task setting, and dataset choice. Without a more general empirical analysis demonstrating robustness across these conditions, I am not convinced by the effectiveness of the proposed method.

---

> > > ### Author Response · Authors · 2026-04-06
> > >
> > > Thanks for the follow-up questions. We clarify the human preference evaluation below. To address the robustness concern more directly, we additionally provide image-generation results across multiple schedulers on a separate DFM backbone. All additional empirical results are provided in  [this anonymous link](https://anonymous.4open.science/r/TR-CIE-Responses-A2EC).
> > >
> > > **(How the “small-scale human preference evaluation” was constructed or conducted.)**
> > >
> > > Thanks for pointing this out. Our previous rebuttal did not describe this protocol clearly enough. At NFE = 4, we randomly sample 100 prompts from the 553 GenEval prompts. For each prompt, we generate one image from each of the five samplers using the same pretrained FUDOKI [1] backbone and the same original probability-path configuration. Three annotators then select one preferred image per prompt based on overall prompt faithfulness and visual quality. We aggregate the resulting 300 votes and report the selection frequency of each method.
> > >
> > > We also provide additional qualitative examples and prompts in Figure 1 of the anonymous link.
> > >
> > > **(More general empirical analysis to demonstrate robustness.)**
> > >
> > > Thanks for the comment. We agree that the schedule/path design is an important factor in this setting.  In the pretrained FUDOKI setup, however, this component is tied to the model’s original training configuration rather than being a purely inference-time choice. Changing it only at inference would therefore confound the sampler effect with a mismatch to the pretrained model setup. For this reason, in the FUDOKI experiments, all samplers are evaluated under the same pretrained backbone and the same probability-path configuration, so that the comparison isolates the sampling method itself.
> > >
> > > Meanwhile, the original implementation of FUDOKI generates images with 384×384 resolution by default, and the paper reports an overall training cost of about 43,000 GPU hours. The model is also adapted to the authors’ collected multimodal dataset. Re-training under different resolutions or datasets would therefore go well beyond the scope of a sampler ablation.
> > >
> > > To address the reviewer’s robustness concern more directly, we additionally conduct image-generation experiments on CIFAR10 (32×32) using the DFM backbone of [2], following its original setting. We train and evaluate under four schedulers: linear, quadratic, cubic, and cosine. We report the FID-versus-NFE results in Figure 2 of the anonymous link.
> > >
> > > Overall, our experiments now cover text generation, image generation, text-to-image generation, and a synthetic countdown task. While this is not an exhaustive study over all resolutions and datasets, it provides additional evidence that the behavior of TR-CIE is not specific to a single task, a single image backbone, or a single schedule choice.
> > >
> > > [1] FUDOKI: Discrete Flow-based Unified Understanding and Generation via Kinetic-Optimal Velocities, NeurIPS 2025.
> > >
> > > [2] Discrete Flow Matching, NeurIPS 2024.

---

### Decision · Program_Chairs · 2026-04-30

**Decision:**

Accept (regular)

**Comment:**

This paper introduces TR-CIE, a novel sampler for discrete flow matching by utilizing prior caches model outputs and time reparametrization. This improves sampling quality in the low NFE regime. The authors also provide theoretical justification regarding the efficacy of one step of their sampler and establish a convergence bound. This is an important direction in the field. The reviewers raised concerns about:
1. Empirical evaluation based on generative perplexity for text diffusion, Lack of human evaluation data for image generation.
2. Runtime comparisons.
3. Theoretical differences between local error and final error (the paper deals with local error).

These were extensively discussed and addressed during the rebuttal. Based on reviewer feedback, I recommend weak accept.